


**A Study of Earthquake Recurrence based on a One-body**
**Spring-slider Model in the Presence of Thermal-pressurized**
**Slip-weakening Friction and Viscosity**
Jeen-Hwa Wang
Institute of Earth Sciences, Academia Sinica, P.O. Box 1-55, Nangang, Taipei,
TAIWAN (e-mail: jhwang@earth.sinica.edu.tw)
**Abstract** Earthquake recurrence is studied from the temporal variation in slip
through numerical simulations based on the normalized form of equation of motion of
a one-body spring-slider model with thermal-pressurized slip-weakening friction and
viscosity. The main parameters are the normalized characteristic displacement, $U_c$, of
the friction law and the normalized damping coefficient (to represent viscosity), $\eta$.
Define $T_R$, $D$, and $\tau_D$ to be the recurrence time of events, the final slip of an event,
and the duration time of an event, respectively. Simulation results show that $T_R$
increases when $U_c$ decreases or $\eta$ increases; $D$ and $\tau_D$ decrease with increasing $\eta$; and
$\tau_D$ increases with $U_c$. The time- and slip-predictable model can describe the temporal
variation in cumulative slip. When the wear process is taken into account, the
thickness of slip zone, $h$ which depends on the cumulated slip, $S(t)=\sum D(t)$, i.e.,
$h(t)=CS(t)$ ($C$=a dimensionless constant) is an important parameter on $T_R$ and $D$. $U_c$ is
a function of $h$ and thus depends on $C$. In the computational time period, the wear
process influences the recurrence of events and such an effect increases with $C$ when





$C>0.0001$. Both $T_R$ and $D$ decrease when the fault becomes more mature, thus
suggesting that it is more difficult to produce large earthquakes along a fault when it
becomes more mature. Neither the time-predictable model nor the slip-predictable one
can describe the temporal variation in $S(t)$ under the wear process with large $C$.
**Key Words**: Recurrence of events, final slip, rise time, one-body spring-slider
model, thermal-pressurized slip-weakening friction, characteristic displacement,
viscosity, wear process

## 1 Introduction

Earthquake recurrence that is relevant to the physics of faulting is an important
factor in seismic hazard assessment. It is related to the seismic cycle, which represents
the occurrence of several earthquakes in the same segment of a fault during a time
period. Fig. 1 exhibits the general pattern of time variation in slip during a seismic
cycle. In the figure, $T_R$ is the recurrence (also denoted by repeat or inter-event) time
of two events in a seismic cycle, $\tau_D$ is the duration time of slip of an event, and $D$ is
the final slip of an event. Sykes and Quittmeyer (1981) pointed out that the major
factors in controlling $T_R$ are the plate moving speed and the geometry of the rupture
zone. Based on Reid's elastic rebound theory (Reid, 1910), Schwartz and
Coppersmith (1984) assumed that an earthquake occurs when the tectonic shear stress
on a fault is higher than a critical level, which is dependent on the physical conditions
of the fault and the loading by regional tectonics. Since in their work a fault has a
homogeneous distribution of physical properties under constant tectonic loading,
earthquakes could happen regularly.



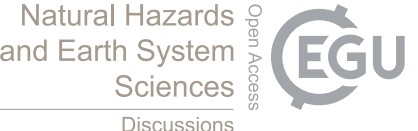

Some observations exhibit periodicity for different size earthquakes. Bakun and
McEvilly (1979) obtained $T_R \approx 23 \pm 9$ years for M$\approx$6 earthquakes at the Parkfield
segment of the San Andreas fault, USA since 1857. Sykes and Menke (2006)
estimated $T_R \approx 100$ years for $M \geq 8$ earthquakes in the Nankaido trough, Japan. Okada et
al. (2003) gained $T_R$=5.5$\pm$0.7 years for earthquakes with $M$=4.8$\pm$0.1 off Kamaishi,
Japan, since 1957. Nadeau and Johnson (1998) inferred an empirical relation between
$T_R$ and seismic moment, $M_o$: $T_R \propto M_o^{1/6}$. To make this relation valid, the stress drop,
$\Delta\sigma$, or the long-term slip velocity of a fault, $v_l$, must be in terms of $M_o$. Based on
three data set from eastern Taiwan, Parkfield, USA, and northeastern Japan, Chen et
al. (2007) inferred $T_R \sim M_o^{0.61}$. Beeler et al. (2001) proposed a theoretical relation:
$T_R = \Delta\sigma^{2/3} M_o^{1/3}/1.81\mu v_l$, where $\mu$ is the rigidity of the fault-zone materials, under
constant $\Delta\sigma$ and $v_l$.
However, the main factors in influencing earthquake occurrences commonly are
spatially heterogeneous and also vary with time. Thus, the recurrence times of
earthquakes, especially large events, are not constant inferred either from observations
(Ando, 1975; Sieh, 1981; Kanamori and Allen, 1986; Wang and Kuo, 1998; Wang,
2005; Sieh et al., 2008) or from modeling (Wang, 1995, 1996; Ward, 1996, 2000;
Wang and Hwang, 2001). Kanamori and Allen (1986) observed that faults with longer
$T_R$ are stronger than those with shorter $T_R$. Davies et al. (1989) proposed that the
longer it has been since the last earthquake, the longer the expected time till the next.
Wang and Kuo (1998) observed that for $M \geq 7$ earthquakes in Taiwan $T_R$ strongly
follows the Poissonian processes. Enescu et al. (2008) found that the distribution of
$T_R$ can be described by an exponential function. From the estimated values of $T_R$ of
earthquakes happened on the Chelungpu fault in central Taiwan from trenching data,
Wang (2005) found that the earthquakes occurred non-periodically.



In order to interpret earthquake recurrences, Shimazaki and Nakata (1980)
proposed three simple phenomenological models. Each model has a constantly
increasing tectonic stress that is controlled by a critical stress level, $\sigma_c$, for failure and
a base stress level, $\sigma_b$. The three models are: (1) the perfectly periodic model (with
constant $\sigma_c$, $\sigma_b$, and $\Delta\sigma$); (2) the time-predictable model (with constant $\sigma_c$, variable
$\sigma_b$, and variable $\Delta\sigma$); and (3) the slip-predictable model (with variable $\sigma_c$, constant $\sigma_b$,
and variable $\Delta\sigma$). For the first model, both $T_R$ and $D$ of next earthquake can be
predicted from the values of $T_R$ or $D$ of previous ones. For the second model, $T_R$ of
next earthquake can be predicted from the values of $D$ of previous ones. For the third
model, $D$ of next earthquake can be predicted from the values of $T_R$ of previous ones.
However, debates about the three models have been made for a long time. Some
examples are given below. Ando (1975) suggested that the second model worked for
post-1707 events, yet not for pre-1707 ones in the Nankaido trough, Japan. Wang
(2005) assumed that the second model could describe the earthquakes occurred on the
Chelungpu fault, Taiwan in the past 1900 years. For the Parkfield earthquake
sequence, Bakun and McEvilly (1984) took different models; while Murray and
Segall (2002) considered the failure of the second model. From laboratory results,
Rubinstein et al. (2012) assumed the failure of the time- and slip-predictable models
for earthquakes.
Some models, for instance the crack model and dynamical spring-slider model,
have been developed for fault dynamics, even though the seismologists have not a
comprehensive model. There are several factors in controlling fault dynamics and
earthquake ruptures (see Bizzarri, 2009; Wang, 2017b). Among the factors, friction
(Nur, 1978; Belardinelli and Belardinelli, 1996) and viscosity (Jeffreys, 1942; Spray,
1993; Wang, 2007) are two significant ones.



Modeling earthquake recurrence based on different models has been long made and
is reviewed by Bizzarri (2012a,b) and Franović et al. (2016). Among the models, the
spring-slider model has been used to study fault dynamics and earthquake physics
(see Wang 2008). Burridge and Knopoff (1967) proposed the one-dimensional
N-body model (abbreviated as the 1-D BK model henceforth). Wang (2000, 2012)
extended the 1-D model to 2-D one. The one-, two-, three-, and few-body models with
various friction laws have also been applied to approach fault dynamics (see Turcotte,
1992). The studies for various friction laws based on spring-slider models are briefly
described below: (1) for rate- and state-dependent friction (e.g., Rice and Tse, 1986;
Ryabov and Ito, 2001; Erickson et al., 2008, 2011; He et al., 2003; Mitsui and
Hirahara, 2009; Bizzarri, 2012a; Abe and Kato, 2013;Kostić et al., 2013a; Bizzarri
and Crupi, 2014; Franović et al., 2016); (2) for velocity-weakening friction (e.g.,
Carlson and Langer, 1989; Huang and Turcotte, 1992; Brun and Gomez, 1994; Wang
and Hwang, 2001; Wang, 2003; Kostić et al., 2013b); (3) for simple static/dynamic
friction (e.g., Abaimov et al., 2007; Hasumi, 2007).
Some results concerning earthquake recurrence are simply explained below.
Erickson et al. (2008) suggested that aperiodicity in earthquake dynamics is due to
either the nonlinear friction law (Huang and Turcotte, 1990) or the heterogeneous
stress distribution (Lapusta and Rice, 2003). Wang and Hwang (2001) emphasized the
importance of heterogeneous frictional strengths. Mitsui and Hirahara (2009) pointed
out the effect of thermal pressurization. Dragoni and Piombo (2011) found that
variable strain rate causes aperiodicity of earthquakes. Bizzarri and Crupi (2014)
found that $T_R$ is dependent on the loading rate, effective normal stress, and
characteristic distance of the rate- and state-dependent friction law.



As mentioned previously, numerous studies have been made for exploring the
frictional effect on earthquake recurrence. But, the study concerning the viscous effect
on earthquake recurrence is rare. In the followings, we will investigate the effects of
slip-weakening friction due to thermal-pressurization and viscosity on earthquake
recurrence based on the one-body spring-slider model.

**2 One-body Model**
Fig. 2 displays the one-body spring-slider model. In the model, $m$, $K$, $N$, $F$, $\eta$, $u$, $v$
($=du/dt$), $v_p$, and $u_o=v_p t$ denote, respectively, the mass of the slider, the stiffness (or
spring constant) of the leaf spring, the normal force, the frictional force between the
slider and the moving plate, the damping coefficient (to represent viscosity as
explained below), the displacement of the slider, the velocity of the slider, the plate
moving speed, and the equilibrium location of the slider. The frictional force $F$ (with
the static value of $F_o$) is usually a function of $u$ or $v$. Viscosity results in the viscous
force, $\Phi$, between the slider and the moving plate, and $\Phi$ is in terms of $v$. A driving
force, $Kv_p t$, caused by the moving plate through the leaf spring pulls the slider to
move. The equation of motion of the model is:

$md^2u/dt^2=-K(u-u_o)-F(u,v)-\Phi(v).$              (1)


When $Kv_p t \geq F_o$, $F$ changes from static frictional force to dynamic one and thus makes
the slider move.
The frictional force $F(u,v)$ is controlled by several factors (see Wang, 2016; and
cited references therein). An effect combined from temperature and fluids in a fault
zone can result in thermal pressurization (abbreviated as TP below) which would yield

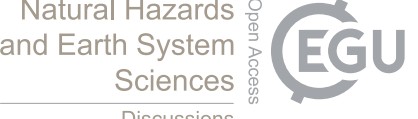



a shear stress (resistance) on the fault plane (Sibson, 1973; Lachenbruch, 1980; Rice,
2006; Wang, 2009, 2011, 2016, 2017a,b,c; Bizzarri, 2009). Rice (2006) proposed two
end-member models of TP, i.e., the adiabatic-undrained-deformation (AUD) model
and slip-on-a-plane (SOP) model. The latter is not appropriate in this study because of
the request of constant velocity. The former is related to a homogeneous simple strain
$\varepsilon$ at a constant normal stress $\sigma_n$ on a spatial scale of the sheared layer. Its shear
stress-slip function, $\tau(u)$, is: $\tau(u)=f(\sigma_n-p_o)exp(-u/u_c)$ (Rice, 2006), which decreases
exponentially with increasing $u$. The characteristic displacement is $u_c=\rho_f C_v h/\mu_f \Lambda$,
where $\rho_f$, $C_v$, $h$, $\mu_f$, and $\Lambda$ are, respectively, the fluid density, heat capacity (in
J/ºC/kg), the thickness, frictional strength, and the undrained pressurization factor of
the fault zone.
Based on the AUD model, Wang (2009, 2016, 2017a,b,c) took a simplified slip-
weakening friction law (denoted by the TP law hereafter):

$F(u)=F_o exp(-u/u_c).$ (2)

An example of the plot of $F(u)$ versus $u$ for five values of $u_c$, i.e., 0.1, 0.3, 0.5, 0.7,
and 0.9 m, which are taken from Wang (2016), is displayed in Fig. 3. $F(u)$ decreases
with increasing $u$ and its decreasing rate, $\gamma$, decreases with increasing $u_c$. The force
drop is lower for larger $u_c$ than for smaller $u_c$. When $u<<u_c$, $exp(-u/u_c)\approx1-u/u_c$, thus
indicating that $u_c^{-1}$ is almost $\gamma$ at small $u$. This TP law is used in this study.
A detailed description about viscosity and the viscous force $\Phi(v)$ can be found in
Wang (2016), and only a brief explanation is given below. Jeffreys (1942) first and
then numerous authors (Byerlee, 1968; Turcotte and Schubert, 1982; Scholz, 1990;
Rice et al., 2001; Wang, 2016) emphasized the viscous effect on faulting due to




174 frictional melts. The viscosity coefficient, $\upsilon$, of rocks is influenced by $T$ (see Turcotte

175 and Schubert, 1982; Wang, 2011). Spray (2005; and cited references therein) observed

176 a decrease in $\upsilon$ with increasing $T$. He also stressed that frictional melts with low $\upsilon$

177 could produce a large volume of melting, thus reducing the effective normal stress.

178 This behaves like fault lubricants during seismic slip.

179  The physical models of viscosity can be found in several articles (e.g., Cohen, 1979;

180 Hudson, 1980). The stress–strain relationship is $\sigma=E\varepsilon$ where $\sigma$ and $E$ are, respectively,

181 the stress and the elastic modulus for an elastic body and $\sigma=\upsilon(d\varepsilon/dt)$, where $\upsilon$ is the

182 viscosity coefficient, for a viscous body. Two simple models with a viscous damper

183 and an elastic spring are often used to describe the viscous materials. A viscous

184 damper and an elastic spring are connected in series leading to the Maxwell model

185 and in parallel resulting in the Kelvin-Voigt model (or the Voigt model). According to

186 Hudson (1980), Wang (2016) proposed that the latter is more suitable than the former

187 for seismological problems and thus the Kelvin-Voigt model, whose constitution law

188 is $\sigma(t)=E\varepsilon(t)+\upsilon d\varepsilon(t)/dt$, is taken here and displayed in Fig. 2. The viscous stress is $\upsilon v$.

189  In order to investigate the viscous effect in a dynamical system, Wang (2016)

190 defined the damping coefficient, $\eta$, based on the phenomenon that an oscillating body

191 damps in viscous fluids. According to Stokes' law, $\eta=6\pi R\upsilon$ for a sphere of radius $R$ in

192 a viscous fluid of $\upsilon$ (see Kittel et al., 1968). Hence, the viscous force in Equation 1 is

193 represented by $\Phi=\eta v$. Note that the unit of $\eta$ is N(m/s)$^{-1}$.

194  Some authors (Knopoff et al., 1973; Cohen, 1979; Rice, 1993; Xu and Knopoff,

195 1994; Knopoff and Ni, 2001; Bizzarri, 2012a; Dragoni and Santini, 2015) considered

196 that viscosity plays a role on causing seismic radiation to release strain energy during

197 faulting.






## 3 Normalization of Equation of Motion


Putting Eq. 2 and $\Phi=\eta v$ into Eq. 1 leads to

$$md^2u/dt^2=-K(u-v_pt)-F_o exp(-u/u_c)-\eta v. \qquad (3)$$

Eq. 3 is normalized for easy numerical computations based on the normalization
parameters, which is dimensionless: $D_o=F_o/K$, $\omega_o=(K/m)^{1/2}$, $\tau=\omega_o t$, $U=u/D_o$, and
$U_c=u_c/D_o$. The normalized velocity, acceleration, and driving velocity are $V=dU/d\tau=$
$[F_o/(mK)^{1/2}]^{-1}du/dt$, $A=d^2U/d\tau^2=(F_o/m)^{-1}d^2u/dt^2$, and $V_p=v_p/(D_o\omega_o)$, respectively.
Define $\Omega=\omega/\omega_o$ to be the dimensionless angular frequency, and thus the phase $\omega t$
becomes $\Omega\tau$. For the purpose of simplification, $\eta/(mK)^{1/2}$ is denoted by $\eta$ below.
Substituting all normalization parameters into Eq. 3 leads to

$$d^2U/d\tau^2=-U-\eta dU/d\tau-exp(-U/U_c)+V_p\tau. \qquad (4)$$

In order to numerically solve Eq. (4), we define two new parameters, i.e., $y_1=U$ and
$y_2=dU/d\tau$. Eq. 4 can be re-written as two first-order differential equations:

$$dy_1/d\tau=y_2 \qquad (5a)$$

$$dy_2/d\tau=-y_1-\eta y_2-exp(-y_1/U_c)+V_p\tau. \qquad (5b)$$

We can numerically solve Eq. 5 by using the fourth-order Runge-Kutta method (Press
et al., 1986). Because of $v_p\approx10^{-10}$ m/s, $V_p$ must be much smaller than 1. To shorten the


223 computational times, $V_p$ is taken to be $10^{-2}$. The backward slip is not allowed in the

224 simulations, because of common behavior of forward earthquake ruptures.

225  A phase portrait, which is a plot of a physical quantity, $y$, versus another, $x$, i.e.,

226 $y=f(x)$, is commonly used to represent nonlinear behavior of a dynamical system

227 (Thompson and Stewart, 1986). The intersection point between $f(x)$ and the bisection

228 line of $y=x$, is defined as the fixed point, that is, $f(x)=x$. If $f(x)$ is continuously

229 differentiable in an open domain near a fixed point $x_f$ and $|f'(x_f)|<1$, attraction can

230 appear at the fixed point. Chaos can also be generated at some attractors. The details

231 can be seen in Thompson and Stewart (1986). In this study, the phase portrait is the

232 plot of $V/V_{max}$ versus $U/U_{max}$.

234 **4 Simulation Results**

235  The results of numerical simulations are shown in Figs. 4–12. The temporal

236 variations in $V/V_{max}$ (displayed by thin solid lines) and cumulative slip $\Sigma U/U_{max}$

237 (displayed by solid lines) are displayed in the left-handed-side panels; while the phase

238 portraits of $V/V_{max}$ versus $U/U_{max}$ (displayed by solid lines) are shown in the right-

239 handed-side panels. Because the maximum values of both $V$ and $U$ decrease from case

240 (a) to case (d), denote the maximum velocity and maximum displacement for case (a)

241 of each figure are denoted by, respectively, $V_{max}$ and $U_{max}$ which are taken as a factor

242 in scaling the waveforms from case (a) to case (d). In the right-handed-side panels of

243 Figs. 4–12, the dashed line represents the bisection line.

244  The cases not including the viscous effect, i.e., $\eta=0$, are first simulated and results

245 are shown in Fig. 4 for four values of $U_c$: (a) for $U_c=0.2$; (b) for $U_c=0.4$; (c) for

246 $U_c=0.8$; and (d) for $U_c=1.0$. The results of the cases including viscosity, i.e., $\eta\neq0$, are

247 displayed in Figs. 5–7 for four values of $\eta$: (a) for $\eta=0.20$; (b) for $\eta=0.40$; (c) for





$\eta$=0.6; and (d) for $\eta$=0.8. The values of $U_c$ are 0.20 in Fig. 5, 0.50 in Fig. 6, and 0.80
in Fig. 7.
Figs. 4–7 show that when $U_c$ and $\eta$ are constants during the computational time
periods, the general patterns of temporal variations in cumulated slip do not change.
Some of the previous studies suggest that the patterns of temporal variations in
cumulated slip can change with time. The changes of $U_c$ and $\eta$ with time should play
the main roles. From $u_c=\rho_f C v h/\mu_f \Lambda$ of the TP model (see Rice 2006), the width of the
slipping zone, $h$, where the maximum deformation is concentrated (Bizzarri, 2009), is
a significant parameter in this study. The reasons to select $h$ to be the main factor are
explained below in the section of "Discussion.". From geological surveys, Rathbun
and Marone (2010) observed that $h$ is not spatially uniform even within a single fault.
Hull (1988) and Marrett and Allmendinger (1990) found that the wear processes
occurring during faulting could widen $h$, and thus $h$ could vary with time. According
to the results gained by several authors (e.g., Power et al., 1988; Robertson, 1983; and
Bizzarri, 2010), Bizzarri (2012b) assumed that $h$ is linearly dependent on the
cumulated slip, $S(t)=\sum D(t)$, and can be represented by $h(t)=CS(t)$ where $C$ is a
dimensionless constant. Since $u_c$ is proportional to $h$ and $U_c=u_c/D_o$, $U_c$ is
proportional to $C$. This means that the more mature the fault is, the thicker its slip
zone is. Simulation results for four values of $C$ are shown in Figs. 8–12: (a) for
$C$=0.0001; (b) for $C$=0.001; (c) for $C$=0.01; and (d) for $C$=0.05 when $U_c$=0.1 and $\eta$=0
in Fig. 8, when $U_c$=0.5 and $\eta$=0 in Fig. 9, when $U_c$=0.9 and $\eta$=0 in Fig. 10, when
$U_c$=0.1 and $\eta$=1 in Fig. 11, and when $U_c$=0.5 and $\eta$=1 in Fig. 12.

**5 Discussion**
The left-handed-side panels in Fig. 4 with $\eta$=0 show that the peak velocity, $V_m$, and



final slip, $D$, with the respective maximum values in case (a) as mentioned above, for
all simulated events decrease with increasing $U_c$. From Fig. 3, the force drop, $\Delta F$,
decreases with increasing $U_c$, thus indicating that larger $\Delta F$ yields higher $V_m$ and
larger $D$. This interprets the negative dependence of $V_m$ and $D$ on $U_c$. The value of $\tau_D$
increases with $U_c$; while $T_R$ decreases with increasing $U_c$. When $U_c=1$, $V_m$ and $D$ are
both very small and the system behaves like creeping of a fault. In the right-handed-
side panels, there are two fixed points: one at $V=0$ and $U=0$ and the other at larger $V$
and larger $U$. The slope values at the two fixed points decrease with increasing $U_c$,
thus suggesting that the fixed point is not an attractor for small $U_c$ and can be an
attractor for larger $U_c$. The phase portrait for $U_c=1$ is very tiny, because the final slip
for $U_c=1.0$ is much smaller than those for $U_c=0.2$, 0.4, and 0.8. Hence, $U_c=1$ will not
be taken into account in the following simulations.

The left-handed-side panels in Figs. 5–7 show that $V_m$ and $D$ decrease when either

$U_c$ or $\eta$ increases; while $\tau_D$ increases with $\eta$ and $U_c$. Meanwhile, $T_R$ increases when
either $\eta$ increases or $U_c$ decreases. The right-handed-side panel exhibits that the
phase portraits are coincided for all simulated events for a certain $\eta$. There are two
fixed points for each case: one at $V=0$ and $U=0$ and the other at larger $V$ and larger $U$.
The slope values at the two fixed points decrease when either $U_c$ or $\eta$ increases. This
suggests that the fixed point is not an attractor for small $U_c$ and low $\eta$, and can be an
attractor for large $U_c$ and high $\eta$. Clearly, the final slip is shorter for $U_c=0.9$ than for
$U_c=0.1$ and 0.5.

From Figs. 5–7, we can see that the temporal variation in cumulative slip can be

described by the perfectly periodic model as mentioned above. Hence, when $U_c$ and $\eta$
do not change with time, the rate of cumulative slip retains a constant in the
computational time period. This is similar to the simulation results with the periodical

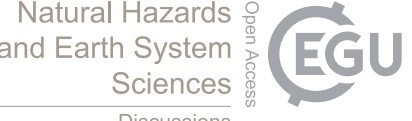



earthquake occurrences obtained by some authors (e.g., Rice and Tse, 1986; Ryabov
and Ito, 2001; Erickson et al., 2008; Mitsui and Hirahara, 2009) based on the
one-body model with rate- and state-dependent friction or velocity- weakening
friction. But, the present result is inconsistent with the simulation results, from which
either the time-predictable model or the slip-predictable model cannot interpret the
temporal variation in cumulative slip, based on the same model obtained by others
(e.g., He et al., 2003; Bazzarri 2012b; Bizzarri and Crupi, 2014; Kostić et al., 2013a,b;
Franović et al., 2016). The differences between the two groups of researchers might
be due to distinct additional constrains in respective studies. Although the detailed
discussion of such differences is important and significant, it is out of the scope of this
study and ignored here.
The phase portraits shown in the right-handed-side panels of Figs. 5–7 exhibit that
the period related to $T_R$ and the size associated with $D$ decrease with increasing h.
This is similar to that obtained from the left-handed-side panels. There are two fixed
points for each case: one at larger $V$ and larger $U$ and the other at $V=0$ and $U=0$. The
slope values of the fixed point at larger $V$ and larger $U$ are higher than 1 and decreases
with increasing $\eta$. This means that larger $\eta$ is easier to generate chaos than small $\eta$.
However, the reducing rate of slop value decreases with increasing $U_c$. The slope
values of the fixed point at $V=0$ and $U=0$ are higher than 1 and only decrease with
increasing $\eta$. This suggests that the fixed points at $V=0$ and $U=0$ can be an attractor.
This behavior becomes weaker when $U_c$ increases.
The previous study demonstrates that when $U_c$ and $\eta$ are constants during the
computational time periods, the general patterns of temporal variations in cumulated
slip cannot change. In order to investigate the effects on the patterns of temporal
variations in cumulated slip, we must consider changes of $U_c$ and $\eta$ with time. The





viscosity coefficient can actually vary immediately before and after the occurrence of
an earthquake (see Spray, 1883, 2005; Wang, 2017b,c). But, a lack of long-term
variation in η does not allow us to explore its possible effect on the change of general
patterns of temporal variations in cumulated slip. Here, only the possible effect due to
time-varying $U_c$.
As mentioned above, $U_c$ is $u_c/D_o$ and thus $U_c = \rho_f C_V h/\mu_f D_o \Lambda$, where $\Lambda =$
$(\lambda_f - \lambda_n)/(\beta_f + \beta_n)$ (Rice, 2006). Obviously, $U_c$ is controlled by six factors, i.e., $\rho_f$, $C_V$, $h$,
$\mu_f$, $D_o$, and $\Lambda$. Since the tectonics of a region is generally stable during a long time,
the value of $D_o = F_o/K$ could not change too much and thus would not influence $U_c$.
The Debye law (cf. Reif, 1965) gives $C_v \sim (T+273.16)^3$, where 273.16 is the value to
convert temperature from Celsius to Kelvin, at low $T$ and $C_v \approx$ constant at high $T$. The
threshold temperature, from which $C_v$ begins to approach a constant, is 200−300 °K.
In this study, $C_v$ is almost a constant because of $T > 250$ °C$= 523.16$ °K, which is the
average ambient temperature of fault zone with depths ranging from 0 to 20 km.
Hence, $C_v$ is almost constant during a long time and thus cannot influence $U_c$.
The frictional strength, $\mu_f$, is influenced by several factors including humidity,
temperature, sliding velocity, strain rate, normal stress, thermally activated rheology
etc (Marone, 1998; Rice, 2006), and thus can change with time (Sibson, 1992; Rice,
2006). Hirose and Bystricky (2007) observed that serpentine dehydration and
subsequent fluid pressurization due to co-seismic frictional heating may reduce $\mu_f$ and
thus promote further weakening in a fault zone. The pore fluid pressure exists in wet
rocks, yet not in dry rocks. Clearly, the time variation in $\mu_f$ can affects the earthquake
recurrences. However, a lack of long-term observations of $\mu_f$ during a seismic cycle
makes the studies of its effect on earthquake recurrence be impossible.
The fluid density $\rho_f$ and the porosity n depend on $T$ and $p$. Although there are





numerous studies on such dependence (Lachenbruch, 1980; Bizzarri, 2012b; and cited
references therein), observed data and theoretical analyses about the values of $\rho_f$ and $n$
during a seismic cycle are rare.

The porosity is associated with the permeability, $\kappa$. Bizzarri (2012c) pointed out

that the time-varying permeability, $\kappa(t)$, and porosity of a fault zone (cf. Mitsui and
Cocco, 2010; Bizzarri, 2012b) can reduce $T_R$. One of the Kozeny–Carman's (KC)
relations (Costa, 2006; and references cited therein) is: $\kappa(t)=\kappa_C\phi^2(t)d^3(t)/[1-\phi(t)]^2$,
where $\kappa_C$ is a dimensionless constant depending on the material in consideration; $\phi$ is
$V_{voids}/V_{tot}$ where $V_{voids}$ and $V_{tot}$ are, respectively, the pore volume and the total volume
of the porous materials; and $d$ is the (average) diameter of the grains, ranging between
$4\times10^{-5}$ m and $1\times10^{-4}$ m (Niemeijer et al., 2010). Usually, $\kappa$, $\phi$, and $d$ can vary in the
fault zone (Segall and Rice, 1995). After faulting $\kappa$ and $\phi$ would change and $d$
becomes smaller because of refining of the grains. According to this relation, Bizzarri
(2012b) found that $\kappa(t)$ could significantly reduce $T_R$ in comparison with the base
model with constant $\kappa$. The reason is explained below. The time-varying permeability
can result in the time-varying pore pressure, $p_f$. This can reduce the frictional
resistance from $\tau=\mu(\sigma_n-p_f)$ and thus could trigger earthquakes earlier. Hence, the
time-varying permeability can change $T_R$. Nevertheless, we cannot investigate its
influence on earthquake recurrence here because there is a lack of a long-term
observation of hydraulic parameters during a seismic cycle.

It is significant to explore the factors that can yield a non-perfectly periodic seismic

cycle. The width of the slipping zone, $h$, can be a candidate as pointed out by some
authors (e.g., Bizzarri, 2009; Rathbun and Marone, 2010). Since the displacement
along a fault is controlled by the fault rheology, $h$ should depend on the rheology on
the sliding interface. The wear processes occurring during faulting could widen $h$



(Hull, 1988; Marrett and Allmendinger, 1990). According to the results gained by
several authors (e.g., Power et al., 1988; Robertson, 1983; and Bizzarri, 2010),
Bizzarri (2012b) proposed a linear dependence of $h$ on the cumulated slip, $S(t)=\sum D(t)$,
i.e., $h(t)=CS(t)$ where $C$ is a dimensionless constant. When the slip zone is thicker, the
fault should be more mature. Since $U_c$ is a function of $h$, $U_c=u_c/D_o=\rho_f C_v h/\mu_f \Lambda D_o$ is
also proportional to $C$ and thus $U_c$ increases with $C$. Hence, numerical simulations for
various values of $C$ and the results are shown in Figs. 8–12, which are different from
Figs. 4–8 especially for large $C$.

The left-handed-side panels of Figs. 8–12 show that $V_m$, $D$, $\tau_D$, and $T_R$ are all

similar when $C \leq 0.001$; while their values are larger for $C=0.01$ than for $C=0.05$ and
also decrease with time. A decrease in $D$ is particularly remarkable when $C \geq 0.01$.
When C=0.05 or $h$ is wider than a critical value, normal earthquakes cannot occur and
only creeping can happen. Obviously, $T_R$ decreases with increasing $C$, thus leading to
an decrease in $T_R$ with increasing $h$. This is similar to the result obtained by Bizzarri
(2010; 2012b). As mentioned above, the fault should be more mature when the slip
zone is thicker. Consequently, both $T_R$ and $D$ decrease when the fault becomes more
mature. This might suggest that it is more difficult to produce large earthquakes along
a fault when it becomes more mature. This implicates that seismic hazard is higher for
a young fault than a mature one. This sounds physically reasonable. Meanwhile,
either the time- or slip- predictable model can only describe the temporal variations of
cumulative slip in the earlier time period, yet not in the later one.

The right-handed-side panels of Figs. 8–12 exhibit that the phase portraits for

$C=0.001$ are slightly different from those for $C=0.0001$ even though the patterns of
their variations in $V$ and $U$ are similar; while the phase portraits for $C>0.001$ are
different from those for $C \leq 0.001$. An increase in $h$ due to an increase in $C$ with



cumulative slip enlarges $U_c$. This can be explained from Fig. 3 which shows that
larger $U_c$ yields a lower $\Delta F$ than smaller $U_c$. Hence, an increase in $U_c$ produces a
decrease in $\Delta F$, thus resulting in low $V_m$ and small $D$. In addition, An increase in $U_c$
makes $exp(-U/U_c)$ approach unity, especially for small $U$, thus reducing the nonlinear
effect caused by TP friction.

Unlike Figs. 4–7, the size of phase portraits in the right-handed-side panels of Figs.

8–12 decreases with increasing $C$. This reflects a decrease in $T_R$ and $D$ of events with
increasing $C$ as mentioned previously. In the phase portraits, there are two fixed
points for each case: one at larger $V$ and larger $U$ and the other at $V=0$ and $U=0$. The
slope values at the fixed point at larger $V$ and larger $U$ are higher than 1 and only
slightly decreases with time when $C \leq 0.01$; while the values remarkably decrease with
time when $C=0.05$. The slope values at the fixed point at $V=0$ and $U=0$ are higher than
1 and only slightly decreases with time when $\underline{C} \leq 0.01$; while those decrease with time
and finally approaches unity when $C=0.05$. Results suggest that the fixed points at
larger $V$ and larger $U$ for all cases in study are not an attractor; and those at $V=0$ and
$U=0$ can evolve to an attractor with time when $C=0.05$. The phenomenon for $C=0.05$
is more remarkable and the evolution is faster for large $U_c$ than for small $U_c$.

**6 Conclusions**

To study the frictional and viscous effects on earthquake recurrence, numerical

simulations of the temporal variations in cumulative slip have been conducted based
on the normalized equation of a one-body model in the presence of thermal-
pressurized slip-weakening friction and viscosity. The major model parameters of
friction and viscosity are represented, respectively, by $U_c$ and $\eta$, where $U_c=u_c/D_o$ is
the normalized characteristic distance and $\eta$ is the normalized damping coefficient.



Numerical simulation of the time variations in $V/V_{max}$ and cumulative slip $\Sigma U/U_{max}$,
the phase portrait of $V/V_{max}$ versus $U/U_{max}$, and the phase portrait of $exp(-U/U_c)$ versus
$U/U_{max}$ are made for various values of $U_c$ and $\eta$.
Results exhibit that both friction and viscosity remarkably affect earthquake
recurrence. The recurrence time, $T_R$, increase when $\eta$ increases or $U_c$ decreases. The
final slip, $D$, and the duration time of slip, $\tau_D$, of an event slightly decrease when $\eta$ or
$\tau_D$ increases and slightly increases with $U_c$. Considering the effect due to wear process,
the thickness of slip zone, $h$ that depends on the cumulated slip, $S(t)=\sum D(t)$, i.e.,
$h(t)=CS(t)$ ($C$=a dimensionless constant), is an important factor in influencing
earthquake recurrences. Because of $U_c=\rho_f C_v h/\mu_f \Lambda D_o$, $U_c$ increases with $C$. When
$C>0.0001$, the wear process influences the recurrence of slip and the effect increases
with $C$. When the slip zone is thicker, the fault should be more mature and $T_R$
increases. Hence, both $T_R$ and $D$ decrease when the fault becomes more mature. This
might suggest that it is more difficult to produce large earthquakes along a fault when
it becomes more mature. The temporal variation in slip cannot be interpreted by the
time-predictable or slip-predictable model when the fault is affected by wear process
with large $C$. In addition, the size of phase portrait of $V/V_{max}$ versus $U/U_{max}$ decreases
with increasing $C$. This reflects a decrease in $T_R$ and $D$ of events with increasing $C$ as
inferred from the temporal variations in cumulative slip.


*Acknowledgments* The study was financially supported by Academia Sinica and
the Ministry of Science and Technology (Grant No.: MOST-106-2116-M-001-005).



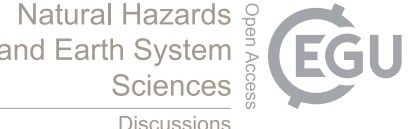

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







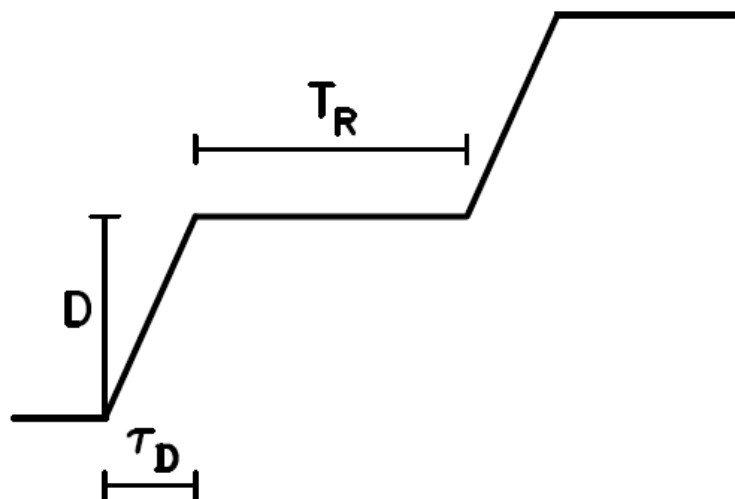


Figure 1. A general pattern of time variation in slip during a seismic cycle: $T_R$=the
recurrence time or the inter-event time of two events in a seismic cycle; $\tau_D$=the
duration time of slip of an event; and $D$=the final slip of an event.







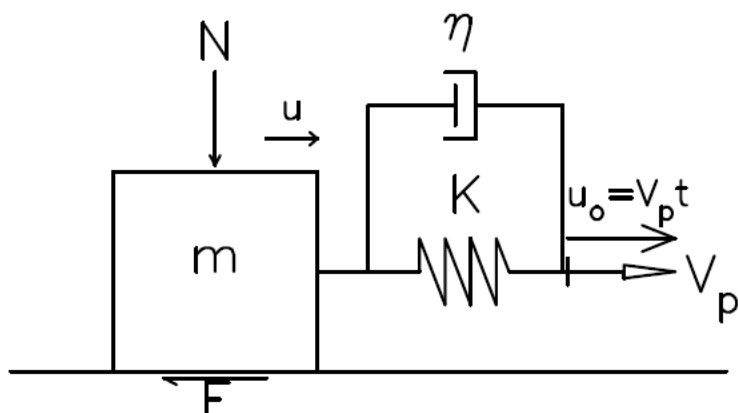

Figure 2. One-body spring-slider model. In the figure, $t$, $m$, $K$, $\eta$, $V_p$, $N$, $F$ $u$, and $u_o$
denote, respectively, the time, the mass of the slider, the spring constant, the
damping coefficient, the driving velocity, the normal force, the frictional force,
displacement of the slider, and the equilibrium location of the slider. (after Wang,
2016)











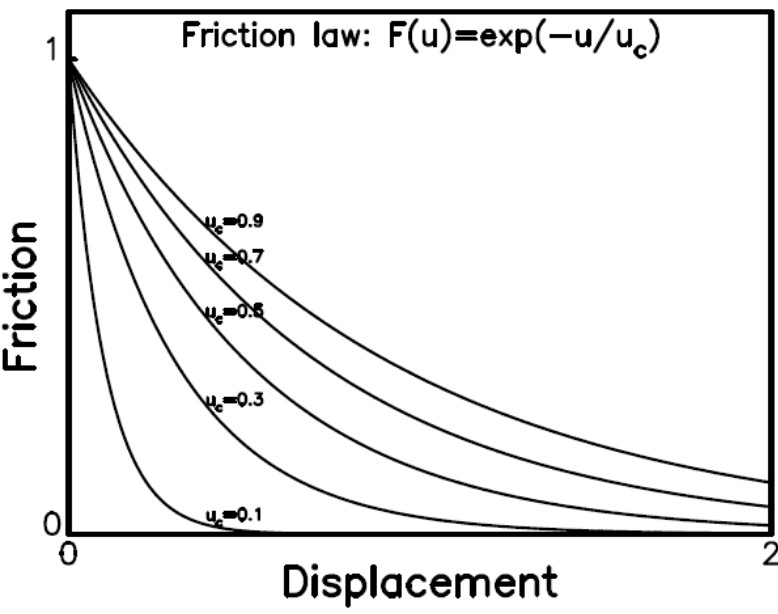


Figure 3. The plots of $F(u)=exp(-u/u_c)$ versus $u$ when $u_c$=0.1, 0.3, 0.5, 0.7, and 0.9 m.
(after Wang, 2016)









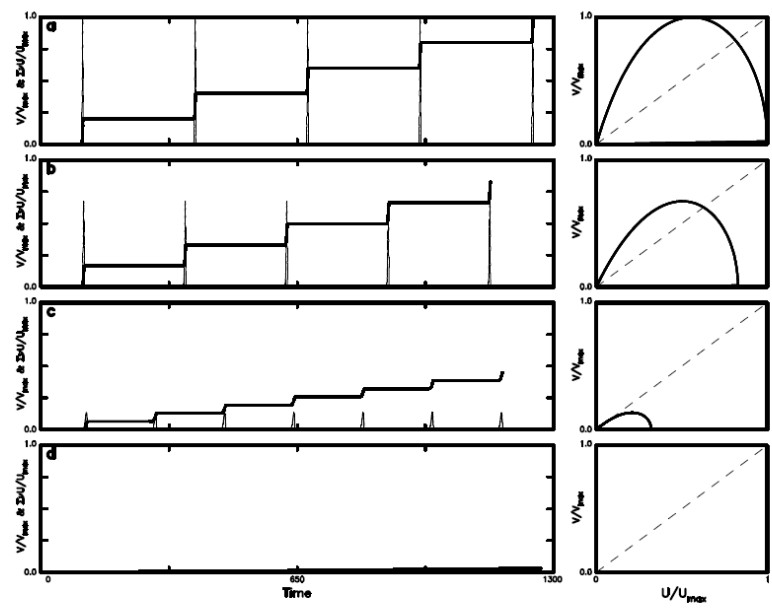


Figure 4. The time variations in $V/V_{max}$ (thin solid line) and cumulative slip $\Sigma U/U_{max}$
(solid line) and the phase portrait of $V/V_{max}$ versus $U/U_{max}$ (solid line) for four
values of $U_c$: (a) for $U_c=0.2$; (b) for $U_c=0.4$; (c) for $U_c=0.8$; and (d) for $U_c=1.0$
when $\eta=0.0$.














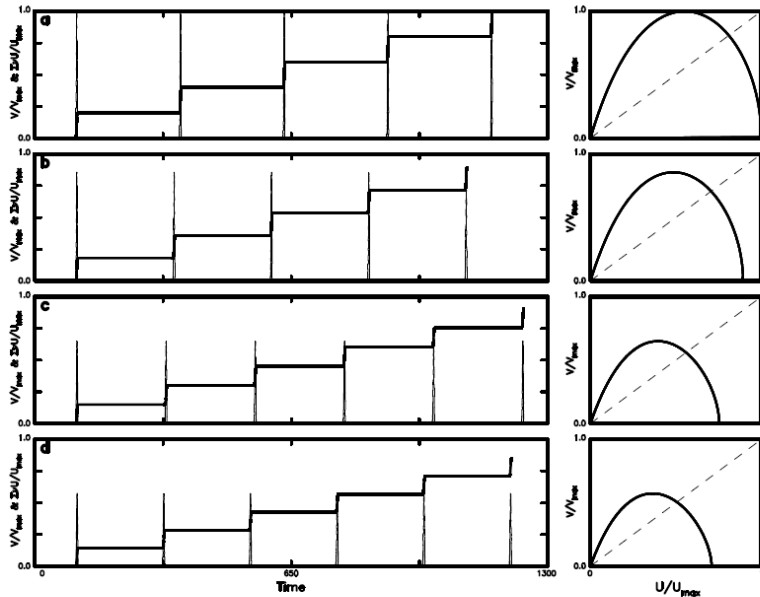

Figure 5. The time variations in $V/V_{max}$ (thin solid line) and cumulative slip $\Sigma U/U_{max}$
(solid line) and the phase portrait of $V/V_{max}$ versus $U/U_{max}$ (solid line) for four
values of $\eta$: (a) for $\eta=0.2$; (b) for $\eta=0.4$; (c) for $\eta=0.8$; and (d) for $\eta=1.0$ when
$U_c=0.2$.








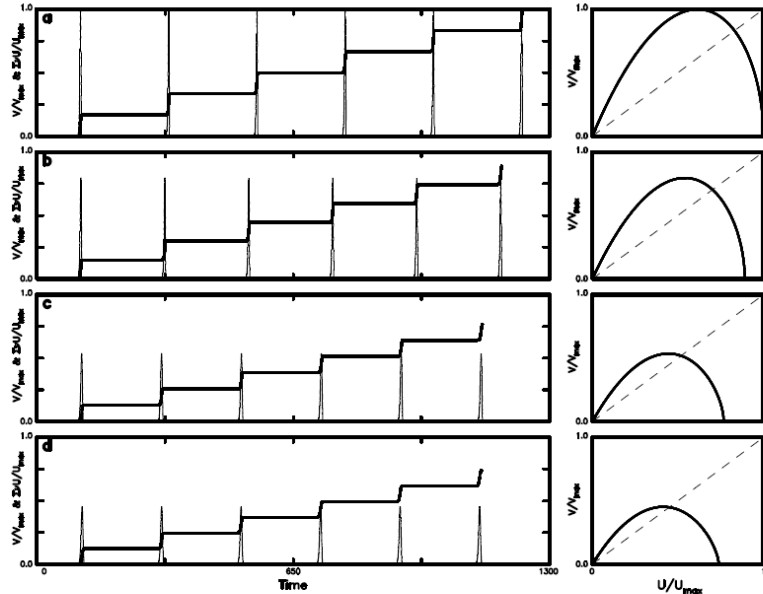


Figure 6. The time variations in $V/V_{max}$ (thin solid line) and cumulative slip $\Sigma U/U_{max}$
(solid line) and the phase portrait of $V/V_{max}$ versus $U/U_{max}$ (solid line) for four
values of $\eta$: (a) for $\eta$=0.2; (b) for $\eta$=0.4; (c) for $\eta$=0.8; and (d) for $\eta$=1.0 when
$U_c$=0.5.








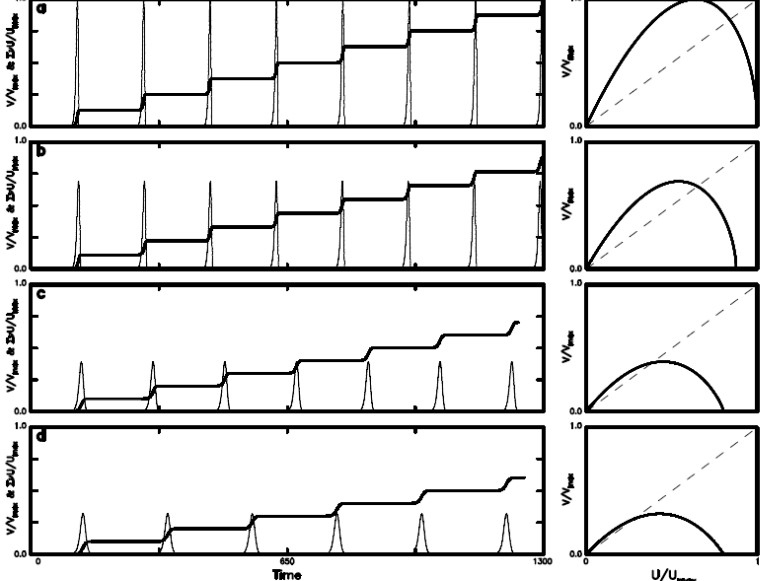

Figure 7. The time variations in $V/V_{max}$ (thin solid line) and cumulative slip $\Sigma U/U_{max}$
(solid line) and the phase portrait of $V/V_{max}$ versus $U/U_{max}$ (solid line) for four
values of $\eta$: (a) for $\eta$=0.2; (b) for $\eta$=0.4; (c) for $\eta$=0.8; and (d) for $\eta$=1.0 when
$U_c$=0.8.









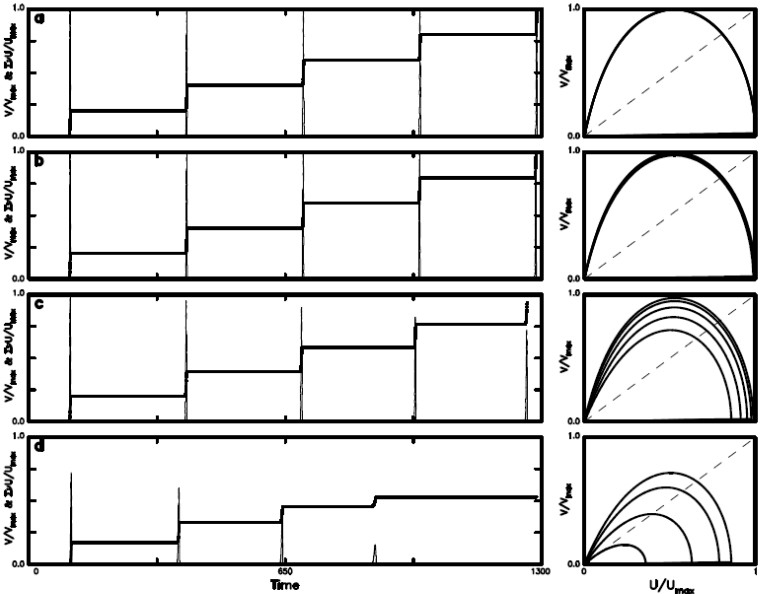


Figure 8. The time variations in $V/V_{max}$ (thin solid line) and cumulative slip $\Sigma U/U_{max}$

(solid line) and the phase portrait of $V/V_{max}$ versus $U/U_{max}$ (solid line) for four

values of $C$: (a) for $C$=0.0001; (b) for $C$=0.001; (c) for $C$=0.01; and (d) for

$C$=0.05 when $U_c$=0.1 and $\eta$=0.








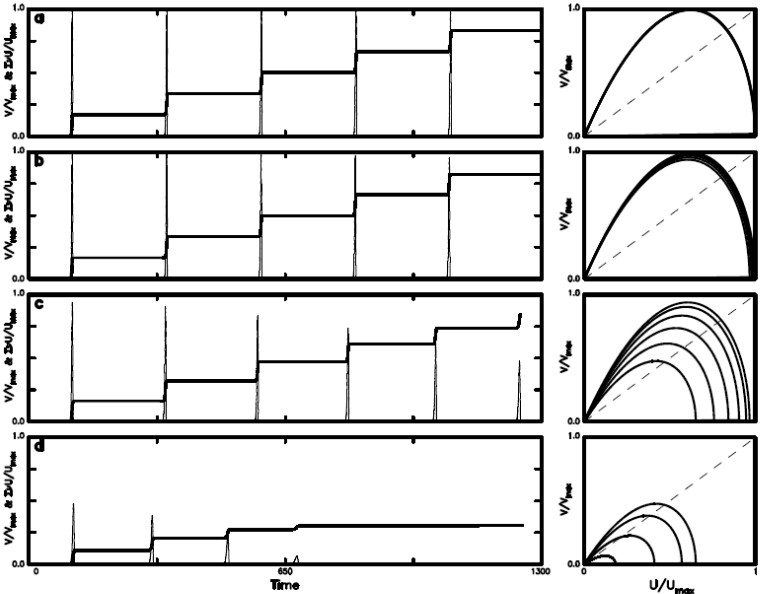


Figure 9. The time variations in $V/V_{max}$ (thin solid line) and cumulative slip $\Sigma U/U_{max}$
(solid line) and the phase portrait of $V/V_{max}$ versus $U/U_{max}$ (solid line) for four
values of $C$: (a) for $C$=0.0001; (b) for $C$=0.001; (c) for $C$=0.01; and (d) for
$C$=0.05 when $U_c$=0.5 and $\eta$=0.








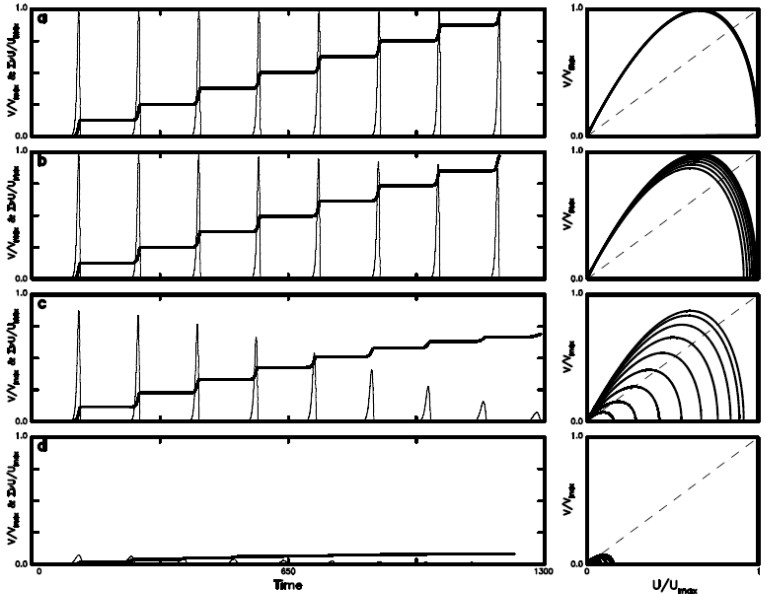


Figure 10. The time variations in $V/V_{max}$ (thin solid line) and cumulative slip $\Sigma U/U_{max}$
(solid line) and the phase portrait of $V/V_{max}$ versus $U/U_{max}$ (solid line) for four
values of $C$: (a) for $C=0.0001$; (b) for $C=0.001$; (c) for $C=0.01$; and (d) for
$C=0.05$ when $U_c=0.9$ and $\eta=0$.








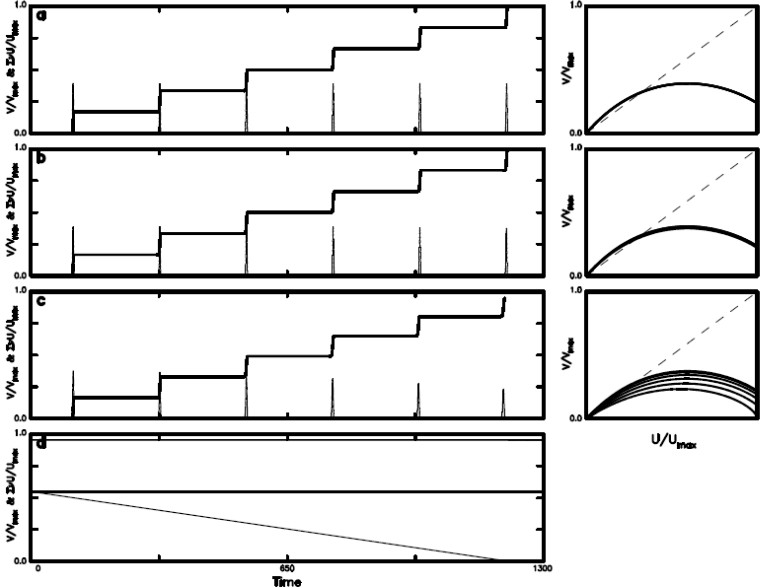


Figure 11. The time variations in $V/V_{max}$ (thin solid line) and cumulative slip $\Sigma U/U_{max}$
(solid line) and the phase portrait of $V/V_{max}$ versus $U/U_{max}$ (solid line) for four
values of $C$: (a) for $C$=0.0001; (b) for $C$=0.001; (c) for $C$=0.01; and (d) for
$C$=0.05 when $U_c$=0.1 and $\eta$=1.






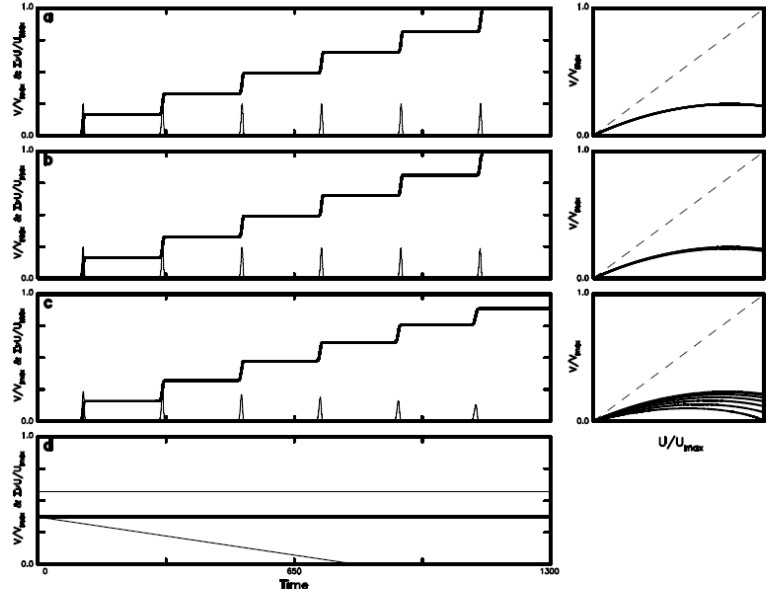

Figure 12. The time variations in $V/V_{max}$ (thin solid line) and cumulative slip $\Sigma U/U_{max}$
(solid line) and the phase portrait of $V/V_{max}$ versus $U/U_{max}$ (solid line) for four
values of $C$: (a) for $C$=0.0001; (b) for $C$=0.001; (c) for $C$=0.01; and (d) for
$C$=0.05 when $U_c$=0.5 and $\eta$=1.