# Peer review of "A Study of Earthquake Recurrence based on a One-body Spring-slider Model in the Presence of Thermal-pressurized Slip-weakening Friction and Viscosity"

_Natural Hazards and Earth System Sciences, 2017_

## Referee Comment (RC1) · Anonymous Referee #1 · 26 Feb 2018

Review of the paper "A study of earthquake recurrence based on a one-body springer-slider model in the presence of thermal-pressurized slip-weakening friction and viscosity" by Jenn-Hwa Wang.

This paper contains a study of earthquake recurrence for a springer-slider model including some kinds of friction and viscosity rheological features. It is well known that the springer-slider model, introduced by Reid more than a century ago, is not able to represent the complex behaviour of real earthquake sources. The careful application of the rheological model does not avoid this flaw of the very simple model. Nevertheless,

this paper offers an extensive review of old and modern literature on the subject, which is worth of being published. The paper seems to me rather clear and written in correct English, but some improvement appears necessary. For instance, the phrase starting with "Because..." at line 239 is quite confusing. In conclusion, I recommend the publication of this paper, with the suggestion of discussing the scarcity of the one-body springer-slider model and a revision of English language.

---

## Author Comment (AC1) · 27 Feb 2018

Response to the comments by Reviewer #1 Thank Reviewer #1 for very valuable comments on my manuscript. My answers to your comments are given below: 1. Among the physical models to approach earthquake faults, the single spring-slider model, which can represent a single fault, is actually the simplest one. However, based on this simple model in the presence of thermal-pressurized friction and viscosity we can obtain good simulations of earthquake recurrences along a single fault. Results can exhibit the frictional and viscous effects on earthquake recurrence.

[Figure]

2. The statements "Because . . . " in Line 239 will be re-written to be "Figures show that the maximum values of both V and U decrease from case (a) to case (d) in each figure. Hence, the maximum velocity and maximum displacement, which are denoted by Vmax and Umax, respectively, for case (a) can be taken as the scaled factor to normalize the waveforms from case (a) to case (d). This makes us easily to compare the waveforms of the four cases in each figure." 3. The statements to show the weak points of the single spring-slider model and the improvement on English writing for the manuscript will be made after I receive the comments by other reviewers.

Please also note the supplement to this comment:
https://www.nat-hazards-earth-syst-sci-discuss.net/nhess-2017-459/nhess-2017-459-AC1-supplement.pdf

---

## Referee Comment (RC2) · Anonymous Referee #2 · 6 Apr 2018

**Review of the paper "A Study of Earthquake Recurrence based on a One-body Spring-slider Model in the Presence of Thermal-pressurized Slip-weakening Friction and Viscosity" by Jeen-Hwa Wang**

This paper studied earthquake recurrence by numerical simulations of a one-degree-of-freedom spring-slider model with thermal-pressurized slip-weakening friction. The paper investigated the effects of the viscosity and the wear process on the recurrence time, slip amount for each event, slip velocity, and so on.

The many parts of the main results stated in the manuscript would not be obtained or read from the simulation results shown in Figs. 4-12. The main reasons of this were the assumption of the constant $U_c$ in the simulations for the examination of the wear effect (Figs. 8-12) and the way of drawing Figs. 4-12.

Regarding the following specific comments [1]-[6] at least, the numerical simulations should be conducted appropriately and the manuscript and figures should be modified before the publication.

**Major comments**

[1] L.23-28 (Abstract), L.385, etc.

The Author stated that the effect of the wear process increases with C. However, the dependency of C on $T_R$ or D is not obtained from the simulation results shown in Figs. 8-12. This is because $U_c$ was assumed to be constant and the same in (a)-(d) for each figure, as stated in L.266-269 and captions of Figs. 8-12, which means that the other parameters (at least one among $\rho_f$, $C_v$, $\mu_f$, $\Lambda$, and $D_0$) varied with C in (a)-(d) for each figure. In order to investigate the effect of C solely, the other parameters ($\rho_f$, $C_v$, $\mu_f$, $\Lambda$, and $D_0$) should be constant and the same in (a)-(d), and thus $U_c$ should change in (a)-(d) and vary with h(t) (i.e., the cumulated slip). It is better to calculate $U_c$ using h=CS(t) for every time step in the simulations.

[2]  ・L.285-286: "The left-handed-side panels in Figs. 5–7 show that $V_m$ and D

decrease when … $U_c$ … increases"

・L.290-293

・L.309-310: "The phase portraits shown in the right-handed-side panels of Figs. 5–

7 exhibit that … the size associated with D decrease with increasing h."

The values of $V_m$, D, and the slope at the two fixed points cannot be compared among Figs. 5-7 because V and U would be normalized by different values of $V_{max}$ and $U_{max}$ among the figures. I guessed that $V_{max}$ and $U_{max}$ correspond to the maximum values of V and U in (a) for each figure and that the maximum values decreases with increasing $U_c$ when $\eta \neq 0$, similar to the cases with $\eta=0$ (Fig. 4).

I suggest that V and U should be normalized by $V_p$ and $V_p\tau_{max}$, respectively, where $\tau_{max}$ is the maximum value of the horizontal axis (1300) in Figs. 4-12.

[3]  ・L.17-18: "$T_R$ increases when $U_c$ decreases or $\eta$ increases"

・L.286-287: "$T_R$ increases when either $\eta$ increases or $U_c$ decreases"

$T_R$ increases when $\eta$ increases for $U_c=0.8$ (Fig. 7), while $T_R$ decreases when $\eta$ increases for smaller $U_c$ (Figs. 5 and 6). The behaviors of stick-slips should be investigated more carefully.

[4]  ・L.276-277: "The value of $\tau_D$ increases with $U_c$"

・L.286: "while $\tau_D$ increases with $\eta$ and $U_c$"

The $\tau_D$ values are unclear in the left panels of Figs. 4-6. Please add the enlarged figures for only one event.

[5] L. 278-282, L.288-292, L.405-407

I cannot understand what "the slope values at the two fixed points" means. $(V/V_{max})/(U/U_{max})$? Or $\frac{U_{max}}{V_{max}}\frac{dV}{dU}$?

[6] Some characters in the numerical formulas are very confusing.

・ About slip and cumulative slip

- u and U in the friction law (equation 2, the second term of the right side of equations 3 and 4, Figure 3, etc.) would represent the time-varying slip amount for one event.

- u and U in $u-u_0$ and $U-V_p\tau$ (the first term of the right side of equations 1, 3, and 4, Figure 1, etc.) show the time-varying cumulative slip.

- Also $\Sigma U$ in Figs.4-12 correspond to the time-varying cumulative slip.

- The (maybe time-varying) cumulative slip used in the wear effect is S(t). Is S(t) the same as $\Sigma U$ in Figs.4-12?

- Is D(t) in $S(t)=\Sigma D(t)$ different from D (final slip of each event, defined at L. 16)?

・ About friction, is $f$ in L.155 the same as $\mu_f$ (L.156 etc.)?

Minor comments

[7] The topic on the wear process starts abruptly at L.20 in Abstract and the last paragraph of Section 4 (Simulation Results, p.11). To clarify the subjects of this paper, it would be better to add the statement that this paper investigated the wear process to the first sentence in Abstract and to the Introduction. In addition, the statements on the wear process in p.11 should be moved to somewhere before Section 4.

[8] L.53: 'the Nankaido trough' → 'the Nankaido segment of the Nankai Trough'?

[9]  L.60: "$T_R = \Delta\sigma^{2/3} M_o^{1/3} / 1.81 \mu v_l$"

The assumption of constant $\Delta\sigma$ and $v_l$ is not needed to derive this relation. If $\Delta\sigma$ or $v_l$ varies with time, also $T_R$ varies with time.

[10] L.71-72

I cannot understand the meaning of 'the distribution of $T_R$'. The probability density distribution of $T_R$?

[11] L.87: 'the Nankaido trough' → 'the Nankai trough'? Or 'the Nankaido segment of the Nankai Trough'?

[12] L.152-153: "The latter is not appropriate in this study because of the request of constant velocity."

The equations of SOP model for variable velocity are shown in Rice (2006), which can be solved numerically. It should be noted that I agree to adopt AUD model in this study in order to examine the wear effect.

[13] L.163 (equation 2)

How did the Author treat equation 2 for the stable sliding (e.g. cases shown in Figs. 11d and 12d)? u=0?

[14] ・L.167-168: "The force drop is lower for larger $u_c$ than for smaller $u_c$."

   ・L.399: "larger $U_c$ yields a lower $\Delta F$ than smaller $U_c$"

The final friction drop is 1, regardless of $u_c$ and $U_c$ (Fig. 3). Did the Author mean "the force drop for a certain displacement"?

[15]p.8

$\upsilon$ decreases with increasing T and $\eta$ is proportional to $\upsilon$. However, $\eta$ was assumed to be constant in this study. I wonder if the simulations with $\eta$ depending on T are possible. The Author does not have to conduct such simulations in this study, but the comments on this may be interesting.

[16]L.222: "$V_p$ must be much smaller than 1"

The value of $V_p$ depends on $D_o\omega_o$. How large is $D_o\omega_o$?

[17]L.223: "$V_p$ is taken to be $10^{-2}$"

Do the results change if $V_p$ is another value?

[18]Section 4 (Simulation Results)

The results of the numerical simulations stated in pp.12-13 and L.381-414 should be moved to Section 4.

[19]L.252-253

The references are needed.

[20]L.264-265, L.377-378 etc.: "$U_c$ is proportional to C"

This phrase seems to be strange for me because the variable is S(t) and C is the proportion coefficient.

[21]L.265: "This"

What does the word "this" show? The sentence just before this word? The fact "the more mature the fault is, the thicker its slip zone is" comes only from h(t)=CS(t).

[22]L.274-276: "the force drop, $\Delta F$, decreases with increasing $U_c$, thus indicating that larger $\Delta F$ yields higher $V_m$ and larger D"

I cannot understand the logic of this sentence. The Author's intention may be "$\Delta F$ decreases with increasing $U_c$ for a certain finite displacement" because the friction drop reaches 1 when displacement is $\infty$ regardless of $U_c$ (Fig.3). If so, however, this phrase have no relation to "larger $\Delta F$ yields larger D".

[23]L.292-293

The $U_c$ values are different from those in L.248 and figure captions.

[24]L.301-305

In a one-degree-of-freedom spring-slider model with constant friction parameters, the system reaches limiting cycles even in the previous studies listed in L.304-305, although I have not checked the results by Kostić et al. (2013a) and Franović et al. (2016). The Author may consider the initial transient phase, but the phase depends on the assumed initial state before the spring starts to be pull with the driving velocity of $V_p$. The behaviors of the limiting cycle reflect the parameters of the friction and of the system properly. It should be noted that the very small transient phase was also observed in Rice and Tse (1986) (the reference in L.298).

[25]L.309-310

I cannot understand that the right panels show $T_R$.

[26]L.314

I cannot understand why larger $\eta$ generates chaos.

[27]L.318

The slope values at V=0 and U=0 decrease with increasing $\eta$ more drastically for the larger $U_c$.

As pointed out in my comment [2], the slope values should not be compared among the figures because $U_{max}$ and $V_{max}$ values are different among the figures.

[28]L.319

The references are needed.

[29]L.321: "the effects"

The effects of temporal variations of $\eta$ and $U_c$?

[30]L.329-330: "$\Lambda=(\lambda_f-\lambda_n)/(\beta_f+\beta_n)$"

It would be better to move this to p.7, adding the definition of $\lambda_f$, $\lambda_n$, $\beta_f$, and $\beta_n$.

[31]L.338: "$\mu_f$" → "$\mu_f$"

[32]L.347: "$\rho_f$" and "n" → "$\rho_f$" and "$n$"

[33]L.362-364

The Author used the words "time-varying". However, "the increase in permeability can result in the increase in pore pressure due to slip" would be better because "This" in the sentence "This can reduce the frictional resistance" obviously means an increase in the pore pressure.

[34]L. 410: "$\underline{C}$" → "$C$"?

[35]L.411: "approaches unity"

The slope values seem to become smaller than unity in Figs. 9-12. Plotting the slope values (with time or slip) may clarify this point.

Why does the unity important? The slope values depend on the $V_{max}$ and $U_{max}$ values.

[36] Bizzarri (2010) showed the effects of the wear process on the stick-slip behaviors, assuming the friction law with thermal pressurization, and thus the results on the wear processes in this study are not new. I suggest that the statements on the results of the simulations including both the wear processes and the viscous effects (Figs. 11 and 12) are added.

[37] Are the $\eta$ and C values used in the simulations consistent with those estimated by observations or laboratory experiments in previous studies (e.g., Boneh et al., 2014, pageoph)?

[38] Vertical aces in Figs. 4-12

Please add the scales of the $\Sigma U/U_{max}$ aces. The maximum of $\Sigma U/U_{max}$ must be larger than 1 because $U/U_{max}$ reaches 1 or larger in the right panels of (a).

[39] Fig.8-12

Why do the behaviors of the stick-slips (e.g., $T_R$, D, and $V_m$) vary with time in spite of the constant $U_c$?

[40] Figs.11(a) and 12(a)

Why $V_m/V_{max} \neq 1$? I guessed that $V_{max}$ was defined as $V_m$ in (a) for each figure in Figs. 4-10.

Why is the maximum of $U/U_{max}$ larger than 1? I guessed that $U_{max}$ was defined as the maximum of U in each figure in Figs. 4-10.

[41] Figs. 11(d) and 12(d)

Why are there two thin solid lines? Why is $\Sigma U/U_{max}$ constant (thick solid line)?

---

## Author Comment (AC2) · 3 May 2018

Response to the comments by Reviewer #1 Thank Reviewer #1 for very valuable comments on my manuscript. The revisions are marked with red. My answers to your comments are given below: 1. Among the physical models to approach earthquake faults, the single spring-slider model, which can represent a single fault, is actually the simplest one. However, based on this simple model in the presence of thermal-pressurized friction and viscosity we can obtain good simulations of earthquake recurrences along a single fault. Results can exhibit the frictional and viscous effects

on earthquake recurrence. 2. The statements "Because ... " in Line 239 will be re-written to be "Figures show that the maximum values of both V and U decrease from case (a) to case (d) in each figure. Hence, the maximum velocity and maximum displacement, which are denoted by Vmax and Umax, respectively, for case (a) can be taken as the scaled factor to normalize the waveforms from case (a) to case (d). This makes us easily to compare the waveforms of the four cases in each figure." 3. The statements to show the weak points of the single spring-slider model and the improvement on English writing for the manuscript will be made after I receive the comments by other reviewers.

Please also note the supplement to this comment:
https://www.nat-hazards-earth-syst-sci-discuss.net/nhess-2017-459/nhess-2017-459-AC2-supplement.pdf

---

## Author Comment (AC3) · 3 May 2018

Response to the Comments by Reviewer #2

Review of the paper "A Study of Earthquake Recurrence based on a One-body Spring-slider Model in the Presence of Thermal-pressurized Slip-weakening Friction and Viscosity" by Jeen-Hwa Wang

This paper studied earthquake recurrence by numerical simulations of a one-degree-of-freedom spring-slider model with thermal-pressurized slip-weakening friction. The

paper investigated the effects of the viscosity and the wear process on the recurrence time, slip amount for each event, slip velocity, and so on. The many parts of the main results stated in the manuscript would not be obtained or read from the simulation results shown in Figs. 4-12. The main reasons of this were the assumption of the constant Uc in the simulations for the examination of the wear effect (Figs. 8-12) and the way of drawing Figs. 4-12. Regarding the following specific comments [1]-[6] at least, the numerical simulations should be conducted appropriately and the manuscript and figures should be modified before the publication. [Answer] I would like to express my thanks to you for carefully reading my manuscript and giving me many valuable comments and suggestions to improve the manuscript. The revisions are marked with red.

Major comments

[1] L.23-28 (Abstract), L.385, etc.: The Author stated that the effect of the wear process increases with C. However, the dependency of C on TR or D is not obtained from the simulation results shown in Figs. 8-12. This is because Uc was assumed to be constant and the same in (a)-(d) for each figure, as stated in L.266-269 and captions of Figs. 8-12, which means that the other parameters (at least one among f, Cv, $\mu$f, $\Lambda$, and D0) varied with C in (a)-(d) for each figure. In order to investigate the effect of C solely, the other parameters (f, Cv, $\mu$f, $\Lambda$, and D0) should be constant and the same in (a)-(d), and thus Uc should change in (a)-(d) and vary with h(t) (i.e., the cumulated slip). It is better to calculate Uc using h=CS(t) for every time step in the simulations. [Answer] Actually, Uc is not constant and varies with h in panels (a)–(d) of Figs. 8–12. The value of Uc written in each figure caption is the initial value, Uco, in the relationship: Uc=Uco+C$\Sigma$U assumed by me. This point has been explained in the revised manuscript. The re-written statements are "Simulation results for four values of C are shown in Figs. 8–12: (a) for C=0.0001; (b) for C=0.001; (c) for C=0.01; and (d) for C=0.05 when ïA̧Í=0 in Figs. 8–10 and when ïA̧Í=1 in Figs. 11–12. The initial values of Uc are 0.1 for Fig. 8, 0.5 for Fig. 9, 0.9 for Fig. 10, 0.1 for Fig. 11, and 0.5 for Fig. 12." Meanwhile, the value

of Uc shown in each figure caption has been replaced by "Uco."

[2] ãČżL.285-286: "The left-handed-side panels in Figs. 5–7 show that Vm and D decrease when . . . Uc . . . increases" ãČżL.290-293 ãČżL.309-310: "The phase portraits shown in the right-handed-side panels of Figs. 5–7 exhibit that . . . the size associated with D decrease with increasing ïĄÍ."

The values of Vm, D, and the slope at the two fixed points cannot be compared among Figs. 5-7 because V and U would be normalized by different values of Vmax and Umax among the figures. I guessed that Vmax and Umax correspond to the maximum values of V and U in (a) for each figure and that the maximum values decreases with increasing Uc when $\eta\neq0$, similar to the cases with $\eta$=0 (Fig. 4). I suggest that V and U should be normalized by Vp and Vp$\tau$max, respectively, where $\tau$max is the maximum value of the horizontal axis (1300) in Figs. 4-12. [Answer] It is very sorry that the notations were not well explained in the original manuscript. The values of Vm and D, respectively, represent the peak value of velocity and final slip of an event in each figure and have been displayed in Fig. 1 which has been re-drawn and different from the original one. The quantities Vmax and Umax are the maximum values of V and U, respectively, in the first panel marked by "a" of a figure with four panels. The normalization scales in the left-handed-side panels of Figs. 4–12 are Vmax for the velocities and the final value of $\Sigma$U/Umax for the cumulative displacements in the computational time. Hence, the upper bound scales are "1" for both the velocity and the displacement. Hence, only the patterns of temporal variations of velocity and cumulative slip are concerned in these figures. The above-mentioned explanations have been added to the revised manuscript

[3] ãČżL.17-18: "TR increases when Uc decreases or $\eta$ increases" ãČżL.286-287: "TR increases when either $\eta$ increases or Uc decreases" TR increases when $\eta$ increases for Uc=0.8 (Fig. 7), while TR decreases when $\eta$ increases for smaller Uc (Figs. 5 and 6). The behaviors of stick-slips should be investigated more carefully. [Answer] The related description has been improved.

[4] ãČżL.276-277: "The value of $\tau$D increases with Uc" ãČżL.286: "while $\tau$D increases with $\eta$ and Uc" The $\tau$D values are unclear in the left panels of Figs. 4-6. Please add the enlarged figures for only one event. [Answer] Figure 1 has been re-drawn to include the temporal variation in particle velocity to meet your request.

[5] L. 278-282, L.288-292, L.405-407: I cannot understand what "the slope values at the two fixed points" means (V/Vmax)/(U/Umax)? Or U_max/V_max dV/dU ? [Answer] The slope means d(V/Vmax)/d(U/Umax)=(Umax/Vmax)(dV/dU). This statement has been added to the revised manuscript. By the way, the term "the slope value" has been replaced by "the absolute value of slope" in the revised manuscript.

[6] Some characters in the numerical formulas are very confusing. About slip and cumulative slip u and U in the friction law (equation 2, the second term of the right side of equations 3 and 4, Figure 3, etc.) would represent the time-varying slip amount for one event. u and U in u-u0 and U-Vp$\tau$ (the first term of the right side of equations 1, 3, and 4, Figure 1, etc.) show the time-varying cumulative slip. Also $\Sigma$U in Figs.4-12 correspond to the time-varying cumulative slip. The (maybe time-varying) cumulative slip used in the wear effect is S(t). Is S(t) the same as $\Sigma$U in Figs.4-12? Is D(t) in S(t)=$\Sigma$D(t) different from D (final slip of each event, defined at L.16)? [Answer] The u and U only represent the time-varying slip and time-varying normalized slip, respectively, for one event and they do not denote the time-varying cumulative slip. The parameter $\Sigma$U represents the time-varying cumulative slip. D(t), which represents the final slip of an event, could be constant in the time history as displayed in Figs. 4–7 when all model parameters do not change with time; while it could vary with time as shown in Figs. 8–12 when one of the model parameters does change with time. Hence, D(t) in S(t)=$\Sigma$D(t) is merely D. This point has been explained in the revised manuscript. About friction, is f in L.155 the same as $\mu$f (L.156 etc.)? [Answer] The "f" in L.155 has been replaced by "ïA■f".

Minor comments

[7] The topic on the wear process starts abruptly at L.20 in Abstract and the last paragraph of Section 4 (Simulation Results, p.11). To clarify the subjects of this paper, it would be better to add the statement that this paper investigated the wear process to the first sentence in Abstract and to the Introduction. In addition, the statements on the wear process in p.11 should be moved to somewhere before Section 4. [Answer] The statement "the wear process" has been moved to the places you suggested in the revised manuscript

[8] L.53: 'the Nankaido trough' →ïĂǎ'the Nankaido segment of the Nankai Trough'? [Answer] The statement has been re-written in the revised manuscript.

[9] "TR=ïĄĎïĄş2/3Mo1/3/1.81$\mu$vl ": The assumption of constant $\Delta\sigma$ and vl is not needed to derive this relation. If $\Delta\sigma$ or vl varies with time, also TR varies with time. [Answer] The related statement has been deleted in the revised manuscript.

[10]L.71-72: I cannot understand the meaning of 'the distribution of TR'. The probability density distribution of TR? [Answer] Yes, you are right. It is the probability density distribution of TR. The related statement has been added in the revised manuscript

[11]L.87: 'the Nankaido trough' →ïĂǎ'the Nankai trough'? Or 'the Nankaido segment of the Nankai Trough'? [Answer] The statement "the Nankaido trough" has been replaced by "the Nankai trough" in the revised manuscript.

[12]L.152-153: "The latter is not appropriate in this study because of the request of constant velocity. " The equations of SOP model for variable velocity are shown in Rice (2006), which can be solved numerically. It should be noted that I agree to adopt AUD model in this study in order to examine the wear effect. [Answer] The statement has been re-written in the revised manuscript.

[13]L.163 (equation 2): How did the Author treat equation 2 for the stable sliding (e.g. cases shown in Figs. 11d and 12d)? u=0? [Answer] The value of F(u) at u=0 if Fo, i.e., the static friction force. The statement has been re-written in the revised manuscript.

[Figure]

[14]ãČżL.167-168: "The force drop is lower for larger uc than for smaller uc." ãČżL.399: "larger Uc yields a lower ∆F than smaller Uc" The final friction drop is 1, regardless of uc and Uc (Fig. 3). Did the Author mean "the force drop for a certain displacement"? [Answer] You are right. The statement "for the same final slip" has been added to the revised manuscript.

[15]p.8: $\upsilon$ decreases with increasing T and $\eta$ is proportional to $\upsilon$. However, $\eta$ was assumed to be constant in this study. I wonder if the simulations with $\eta$ depending on T are possible. The Author does not have to conduct such simulations in this study, but the comments on this may be interesting. [Answer] The statements "Since ïAţ decreases with increasing T, ïAĺ decreases with increasing T. Hence, ïAĺ can vary with time during faulting. This point has been studied by Wang (2017b) for the generation of nuclear phase before an earthquake ruptures. In this study, constant ïAĺ is considered for each case" have been added to the revised manuscript.

[16]L.222: "Vp must be much smaller than 1": The value of Vp depends on Do$\omega$o. How large is Do$\omega$o? [Answer] In this study, is considered to be about 1 m/s.

[17]L.223: "Vp is taken to be 10-2" Do the results change if Vp is another value? [Answer] The statements "Since the value of Vp can only influence the recurrence time, TR, between two events and cannot influence the pattern of time variations in velocities and displacements of events. In order to study earthquake recurrence, there must be numerous modelled events with clear and visualized time functions of displacements and velocities for an event in the computational time period. If Vp=10-10 is considered, TR is very long and thus ïAťD is much shorter than TR. This makes the time function of an event displayed in the variation in slip looks like a step function for the displacements and an impulse for the velocities. Hence, VpïĂ¡10-2 is taken in this study."

[18]Section 4 (Simulation Results): The results of the numerical simulations stated in pp.12-13 and L.381-414 should be moved to Section 4. [Answer] The related statements shown in the section of "Discussion" have been moved to the section of "Simulation Results".

[19]L.252-253: The references are needed. [Answer] The related references have been added to the revised manuscript.

[20]L.264-265, L.377-378 etc.: "Uc is proportional to C": This phrase seems to be strange for me because the variable is S(t) and C is the proportion coefficient. [Answer] The statements have been re-written in the revised manuscript.

[21]L.265: "This": What does the word "this" show? The sentence just before this word? The fact "the more mature the fault is, the thicker its slip zone is" comes only from h(t)=CS(t). [Answer] The sentence has been re-written as "Based on h(t)=CS(t), the more mature the fault is, the thicker its slip zone is." in the revised manuscript.

[22]L.274-276: "the force drop, $\Delta F$, decreases with increasing Uc, thus indicating that larger $\Delta F$ yields higher Vm and larger D" I cannot understand the logic of this sentence. The Author's intention may be "$\Delta F$ decreases with increasing Uc for a certain finite displacement" because the friction drop reaches 1 when displacement is $\infty$ regardless of Uc (Fig.3). If so, however, this phrase have no relation to "larger $\Delta F$ yields larger D". [Answer] The statement "" has been behind the sentence in the revised manuscript.

[23]L.292-293: The Uc values are different from those in L.248 and figure captions. [Answer] The values of Uc shown in L.292-293 are wrong and must the same as those in L.248. The original sentence has been re-written to be "Clearly, like Fig. 4 the final slip decreases with increasing Uc."

[24]L.301-305: In a one-degree-of-freedom spring-slider model with constant friction parameters, the system reaches limiting cycles even in the previous studies listed in L.304-305, although I have not checked the results by KosticÌĄ et al. (2013a) and FranovicÌĄ et al. (2016). The Author may consider the initial transient phase, but the phase depends on the assumed initial state before the spring starts to be pull with the driving velocity of Vp. The behaviors of the limiting cycle reflect the parameters of the

friction and of the system properly. It should be noted that the very small transient phase was also observed in Rice and Tse (1986) (the reference in L.298). [Answer] Your viewpoint is correct. In this study, I mainly focus on the effect on recurrence. The phase portrait is just used to express the possible change of fixed points due to either a use of different values of or a use of time-varying values of model parameters. Nonlinear behavior, including very small transient phase which was not observed in this study, of the system will be my next study.

[25]L.309-310: I cannot understand that the right panels show TR. [Answer] The related statements have been deleted in the revised manuscript.

[26]L.314: I cannot understand why larger $\eta$ generates chaos. [Answer] The word "chaos" has been re-written as "an attractor" in the revised manuscript.

[27]L.318: The slope values at V=0 and U=0 decrease with increasing $\eta$ more drastically for the larger Uc. As pointed out in my comment [2], the slope values should not be compared among the figures because Umax and Vmax values are different among the figures. [Answer] As mentioned in my answer of your comment [2], for a certain figure we can the absolute values of slope in the four right-handed-sides panels because their values of Vmax and Umax are the same. Of course, it is not good to compare the values in different figures due to different values of Vmax and Umax in use.

[28]L.319: The references are needed. [Answer] "The previous study" means the simulation results of this study. Hence, the words "The previous study" have been re-written as "The simulation results as mentioned previously".

[29]L.321: "the effects" The effects of temporal variations of $\eta$ and Uc? [Answer] The following statement "the effects of time-dependent $\eta$ and Uc" has been added to the revised manuscript.

[30]L.329-330: "$\Lambda = (\lambda f - \lambda n)/(\beta f + \beta n)$" It would be better to move this to p.7, adding the definition of $\lambda f$, $\lambda n$, $\beta f$, and $\beta n$. [Answer] The statements have been re-written and

added in the revised manuscript.

[31]L.338: "$\mu$f" →ïĂă"$\mu$f" [Answer] "$\mu$f" is replaced by "ïA■f"ïĂă in the revised manuscript.

[32]L.347: "f" and "n" →ïĂă"f" and "n" [Answer] "f" and "n" are replaced, respectively, byïĂă"f" and "n" in the revised manuscript.

[33]L.362-364: The Author used the words "time-varying". However, "the increase in permeability can result in the increase in pore pressure due to slip" would be better because "This" in the sentence "This can reduce the frictional resistance" obviously means an increase in the pore pressure. [Answer] The statement "The time-varying permeability can result in the time-varying pore pressure, pf" has been re-written as "An increase in permeability can result in an increase in pore pressure, pf".

[34]L. 410: "C" →ïĂă"C"? [Answer] "C" is replaced by "C" in the revised manuscript.

[35]L.411: "approaches unity" The slope values seem to become smaller than unity in Figs. 9-12. Plotting the slope values (with time or slip) may clarify this point. Why does the unity important? The slope values depend on the Vmax and Umax values. [Answer] Actually, the slope values become smaller than unity in Figs. 9-12. Hence, the statement "approach to unity" has been deleted in the revised manuscript.

[36]Bizzarri (2010) showed the effects of the wear process on the stick-slip behaviors, assuming the friction law with thermal pressurization, and thus the results on the wear processes in this study are not new. I suggest that the statements on the results of the simulations including both the wear processes and the viscous effects (Figs. 11 and 12) are added. [Answer] I agree with you. Related information has been added to the revised manuscript.

[37]Are the $\eta$ and C values used in the simulations consistent with those estimated by observations or laboratory experiments in previous studies (e.g., Boneh et al., 2014, pageoph)? [Answer] This study is merely my first step to theoretically explore the

earthquake recurrences caused by time-varying model parameters through numerical simulations. In this study, I just want to theoretical explore the possible effects on earthquake recurrences caused by time-varying of fault width. Hence, only the assumed values have been taken into account. I have not yet compared my values with those obtained by others. I will approach the problem for real faults in near future, and thus it is necessary to take the values of model parameters obtained from field observations and laboratory experiments into account.

[38]Vertical aces in Figs. 4-12: Please add the scales of the $\Sigma$U/Umax aces. The maximum of $\Sigma$U/Umax must be larger than 1 because U/Umax reaches 1 or larger in the right panels of (a). [Answer] In Figs. 8–12, the velocity waveforms and displacements are normalized by the maximum values of each figure. Hence, the upper bound value of the vertical axis is 1. The statements have been added to the revised manuscript.

[39]Fig.8-12: Why do the behaviors of the stick-slips (e.g., TR, D, and Vm) vary with time in spite of the constant Uc? [Answer] The values of Uc are not constant and vary with time in Figs. 8–12. The values of Uc shown in the text and figure captions of have been re-written to be the initial values of Uc.

[40]Figs.11(a) and 12(a): Why Vm/Vmax$\neq$1? I guessed that Vmax was defined as Vm in (a) for each figure in Figs. 4-10. Why is the maximum of U/Umax larger than 1? I guessed that Umax was defined as the maximum of U in each figure in Figs. 4-10. [Answer] It is very sorry that the numerical computation cannot work when C is larger than a certain value which depends on Uc. The normalization of original Fig. 11 and Fig 12 was made based on the maximum values of panel (d), which are wrong. The re-computed results are displayed in the revised manuscript.

[41]Figs. 11(d) and 12(d): Why are there two thin solid lines? Why is $\Sigma$U/Umax constant (thick solid line)? [Answer] It is very sorry that the numerical computation cannot work when C is larger than a certain value. Originally, I kept the bad simulation results for C=0.05 in Figs. 11–12 with ïĄÍ=1, because I wanted to retain the same four

values of C for Figs. 8–12. This idea sounds not good. Hence, I have re-done the numerical computations to find out the upper bound value of C for a certain Uc with ïĄÍ=1. The re-computed results are displayed in Figs. 11–12 of the revised manuscript.

Please also note the supplement to this comment:
https://www.nat-hazards-earth-syst-sci-discuss.net/nhess-2017-459/nhess-2017-459-AC3-supplement.pdf

**Supplement:**

Response to the Comments by Reviewer #2

**Review of the paper "A Study of Earthquake Recurrence based on a One-body Spring-slider Model in the Presence of Thermal-pressurized Slip-weakening Friction and Viscosity" by Jeen-Hwa Wang**

This paper studied earthquake recurrence by numerical simulations of a one-degree-of-freedom spring-slider model with thermal-pressurized slip-weakening friction. The paper investigated the effects of the viscosity and the wear process on the recurrence time, slip amount for each event, slip velocity, and so on.

The many parts of the main results stated in the manuscript would not be obtained or read from the simulation results shown in Figs. 4-12. The main reasons of this were the assumption of the constant $U_c$ in the simulations for the examination of the wear effect (Figs. 8-12) and the way of drawing Figs. 4-12.

Regarding the following specific comments [1]-[6] at least, the numerical simulations should be conducted appropriately and the manuscript and figures should be modified before the publication.

**[Answer] I would like to express my thanks to you for carefully reading my manuscript and giving me many valuable comments and suggestions to improve the manuscript. The revisions are marked with red.**

**Major comments**

[1] L.23-28 (Abstract), L.385, etc.: The Author stated that the effect of the wear process increases with C. However, the dependency of C on $T_R$ or D is not obtained from the simulation results shown in Figs. 8-12. This is because $U_c$ was assumed to be constant and the same in (a)-(d) for each figure, as stated in L.266-269 and captions of Figs. 8-12, which means that the other parameters (at least one among $\rho_f$, $C_v$, $\mu_f$, $\Lambda$, and $D_0$) varied with C in (a)-(d) for each figure. In order to investigate the effect of C solely, the other parameters ($\rho_f$, $C_v$, $\mu_f$, $\Lambda$, and $D_0$) should be constant and the same in (a)-(d), and thus $U_c$ should change in (a)-(d) and vary with h(t) (i.e., the cumulated slip). It is better to calculate $U_c$ using h=CS(t) for every time step in the simulations.

**[Answer] Actually, $U_c$ is not constant and varies with h in panels (a)–(d) of Figs. 8–12. The value of $U_c$ written in each figure caption is the initial value, $U_{co}$, in the relationship: $U_c=U_{co}+C\Sigma U$ assumed by me. This point has been explained in the revised manuscript. The re-written statements are "Simulation results for four values of C are shown in Figs. 8–12: (a) for C=0.0001; (b) for C=0.001; (c) for C=0.01; and (d) for C=0.05 when η=0 in Figs. 8–10 and when η=1 in Figs. 11–12. The initial values of $U_c$ are 0.1 for Fig. 8, 0.5 for Fig. 9, 0.9 for Fig. 10, 0.1 for Fig. 11, and 0.5 for Fig. 12." Meanwhile, the value of $U_c$ shown in each figure caption has been replaced by "$U_{co}$."**

[2] ・L.285-286: "The left-handed-side panels in Figs. 5–7 show that $V_m$ and D decrease when … $U_c$ … increases"

・L.290-293

・L.309-310: "The phase portraits shown in the right-handed-side panels of Figs. 5–7 exhibit that … the size associated with D decrease with increasing η."

The values of $V_m$, D, and the slope at the two fixed points cannot be compared among Figs. 5-7 because V and U would be normalized by different values of $V_{max}$ and $U_{max}$ among the figures. I guessed that $V_{max}$ and $U_{max}$ correspond to the maximum values of V and U in (a) for each figure and that the maximum values decreases with increasing $U_c$ when $\eta \neq 0$, similar to the cases with $\eta=0$ (Fig. 4). I suggest that V and U should be normalized by $V_p$ and $V_p \tau_{max}$, respectively, where $\tau_{max}$ is the maximum value of the horizontal axis (1300) in Figs. 4-12.

**[Answer] It is very sorry that the notations were not well explained in the original manuscript. The values of $V_m$ and D, respectively, represent the peak value of velocity and final slip of an event in each figure and have been displayed in Fig. 1 which has been re-drawn and different from the original one. The quantities $V_{max}$ and $U_{max}$ are the maximum values of V and U, respectively, in the first panel marked by "a" of a figure with four panels. The normalization scales in the left-handed-side panels of Figs. 4–12 are $V_{max}$ for the velocities and the final value of $\Sigma U/U_{max}$ for the cumulative displacements in the computational time. Hence, the upper bound scales are "1" for both the velocity and the displacement. Hence, only the patterns of temporal variations of velocity and cumulative slip are concerned in these figures. The above-mentioned explanations have been added to the revised manuscript**

[3]  ・L.17-18: "$T_R$ increases when $U_c$ decreases or $\eta$ increases"

   ・L.286-287: "$T_R$ increases when either $\eta$ increases or $U_c$ decreases"

$T_R$ increases when $\eta$ increases for $U_c=0.8$ (Fig. 7), while $T_R$ decreases when $\eta$ increases for smaller $U_c$ (Figs. 5 and 6). The behaviors of stick-slips should be investigated more carefully.

**[Answer] The related description has been improved.**

[4]  ・L.276-277: "The value of $\tau_D$ increases with $U_c$"

   ・L.286: "while $\tau_D$ increases with $\eta$ and $U_c$"

The $\tau_D$ values are unclear in the left panels of Figs. 4-6. Please add the enlarged figures for only one event.

**[Answer] Figure 1 has been re-drawn to include the temporal variation in particle velocity to meet your request.**

[5]  L. 278-282, L.288-292, L.405-407: I cannot understand what "the slope values  at the two fixed points" means

$(V/V_{max})/(U/U_{max})$? Or $\dfrac{U_{max}}{V_{max}}\dfrac{dV}{dU}$ ?

**[Answer] The slope means $d(V/V_{max})/d(U/U_{max})=(U_{max}/V_{max})(dV/dU)$. This statement has been added to the revised manuscript. By the way, the term "the slope value" has been replaced by "the absolute value of slope" in the revised manuscript.**

[6]  Some characters in the numerical formulas are very confusing.
- About slip and cumulative slip
- u and U in the friction law (equation 2, the second term of the right side of equations 3 and 4, Figure 3, etc.) would represent the time-varying slip amount for one event.
- u and U in u-$u_0$ and U-$V_p\tau$ (the first term of the right side of equations 1, 3, and 4, Figure 1, etc.) show the time-varying cumulative slip.
- Also $\Sigma U$ in Figs.4-12 correspond to the time-varying cumulative slip.

- The (maybe time-varying) cumulative slip used in the wear effect is S(t). Is S(t) the same as $\Sigma U$ in Figs.4-12?
- Is D(t) in S(t)=$\Sigma$D(t) different from D (final slip of each event, defined at L.16)?
**[Answer] The u and U only represent the time-varying slip and time-varying normalized slip, respectively, for one event and they do not denote the time-varying cumulative slip. The parameter $\Sigma U$ represents the time-varying cumulative slip. D(t), which represents the final slip of an event, could be constant in the time history as displayed in Figs. 4–7 when all model parameters do not change with time; while it could vary with time as shown in Figs. 8–12 when one of the model parameters does change with time. Hence, D(t) in S(t)=$\Sigma$D(t) is merely D. This point has been explained in the revised manuscript.**

- About friction, is $f$ in L.155 the same as $\mu_f$ (L.156 etc.)?
**[Answer] The "$f$" in L.155 has been replaced by "$\mu_f$".**

**Minor comments**

[7] The topic on the wear process starts abruptly at L.20 in Abstract and the last paragraph of Section 4 (Simulation Results, p.11). To clarify the subjects of this paper, it would be better to add the statement that this paper investigated the wear process to the first sentence in Abstract and to the Introduction. In addition, the statements on the wear process in p.11 should be moved to somewhere before Section 4.
**[Answer] The statement "the wear process" has been moved to the places you suggested in the revised manuscript**

[8] L.53: 'the Nankaido trough' → 'the Nankaido segment of the Nankai Trough'?
**[Answer] The statement has been re-written in the revised manuscript.**

[9] "$T_R=\Delta\sigma^{2/3}M_o^{1/3}/1.81\mu v_l$": The assumption of constant $\Delta\sigma$ and $v_l$ is not needed to derive this relation. If $\Delta\sigma$ or $v_l$ varies with time, also $T_R$ varies with time.
**[Answer] The related statement has been deleted in the revised manuscript.**

[10]L.71-72: I cannot understand the meaning of 'the distribution of $T_R$'. The probability density distribution of $T_R$?
**[Answer] Yes, you are right. It is the probability density distribution of $T_R$. The related statement has been added in the revised manuscript**

[11]L.87: 'the Nankaido trough' → 'the Nankai trough'? Or 'the Nankaido segment of the Nankai Trough'?
**[Answer] The statement "the Nankaido trough" has been replaced by "the Nankai trough" in the revised manuscript.**

[12]L.152-153: "The latter is not appropriate in this study because of the request of constant velocity." The equations of SOP model for variable velocity are shown in Rice (2006), which can be solved numerically. It should be noted that I agree to adopt AUD model in this study in order to examine the wear effect.
**[Answer] The statement has been re-written in the revised manuscript.**

[13]L.163 (equation 2): How did the Author treat equation 2 for the stable sliding (e.g. cases shown in Figs. 11d and 12d)? u=0?
**[Answer] The value of F(u) at u=0 if $F_o$, i.e., the static friction force. The statement has been re-written in the revised manuscript.**

[14] · L.167-168: "The force drop is lower for larger $u_c$ than for smaller $u_c$."

· L.399: "larger $U_c$ yields a lower $\Delta F$ than smaller $U_c$"

The final friction drop is 1, regardless of $u_c$ and $U_c$ (Fig. 3). Did the Author mean "the force drop for a certain displacement"?
**[Answer] You are right. The statement "for the same final slip" has been added to the revised manuscript.**

[15]p.8: $\upsilon$ decreases with increasing T and $\eta$ is proportional to $\upsilon$. However, $\eta$ was assumed to be constant in this study. I wonder if the simulations with $\eta$ depending on T are possible. The Author does not have to conduct such simulations in this study, but the comments on this may be interesting.
**[Answer] The statements "Since $\upsilon$ decreases with increasing T, $\eta$ decreases with increasing $T$. Hence, $\eta$ can vary with time during faulting. This point has been studied by Wang (2017b) for the generation of nuclear phase before an earthquake ruptures. In this study, constant $\eta$ is considered for each case" have been added to the revised manuscript.**

[16]L.222: "$V_p$ must be much smaller than 1": The value of $V_p$ depends on $D_o\omega_o$. How large is $D_o\omega_o$?
**[Answer] In this study, is considered to be about 1 m/s.**

[17]L.223: "$V_p$ is taken to be $10^{-2}$" Do the results change if $V_p$ is another value?
**[Answer] The statements "Since the value of $V_p$ can only influence the recurrence time, $T_R$, between two events and cannot influence the pattern of time variations in velocities and displacements of events. In order to study earthquake recurrence, there must be numerous modelled events with clear and visualized time functions of displacements and velocities for an event in the computational time period. If $V_p=10^{-10}$ is considered, $T_R$ is very long and thus $\tau_D$ is much shorter than $T_R$. This makes the time function of an event displayed in the variation in slip looks like a step function for the displacements and an impulse for the velocities. Hence, $V_p=10^{-2}$ is taken in this study."**

[18]Section 4 (Simulation Results): The results of the numerical simulations stated in pp.12-13 and L.381-414 should be moved to Section 4.
**[Answer] The related statements shown in the section of "Discussion" have been moved to the section of "Simulation Results".**

[19]L.252-253: The references are needed.
**[Answer] The related references have been added to the revised manuscript.**

[20]L.264-265, L.377-378 etc.: "$U_c$ is proportional to C": This phrase seems to be strange for me because the variable is S(t) and C is the proportion coefficient.
**[Answer] The statements have been re-written in the revised manuscript.**

[21]L.265: "This": What does the word "this" show? The sentence just before this word? The fact "the more mature the fault is, the thicker its slip zone is" comes only from h(t)=CS(t).
**[Answer] The sentence has been re-written as "Based on $h(t)=CS(t)$, the more mature the fault is, the thicker its slip zone is." in the revised manuscript.**

[22]L.274-276: "the force drop, $\Delta F$, decreases with increasing $U_c$, thus indicating that larger $\Delta F$ yields higher $V_m$ and larger D" I cannot understand the logic of this sentence. The Author's intention may be "$\Delta F$ decreases with increasing $U_c$ for a certain finite displacement" because the friction drop reaches 1 when displacement is $\infty$ regardless of $U_c$ (Fig.3). If so, however, this phrase have no relation to "larger $\Delta F$ yields larger D".
**[Answer] The statement "" has been behind the sentence in the revised manuscript.**

[23]L.292-293: The $U_c$ values are different from those in L.248 and figure captions.
**[Answer] The values of Uc shown in L.292-293 are wrong and must the same as those in L.248. The original sentence has been re-written to be "Clearly, like Fig. 4 the final slip decreases with increasing $U_c$."**

[24]L.301-305: In a one-degree-of-freedom spring-slider model with constant friction parameters, the system reaches limiting cycles even in the previous studies listed in L.304-305, although I have not checked the results by Kostić et al. (2013a) and Franović et al. (2016). The Author may consider the initial transient phase, but the phase depends on the assumed initial state before the spring starts to be pull with the driving velocity of $V_p$. The behaviors of the limiting cycle reflect the parameters of the friction and of the system properly. It should be noted that the very small transient phase was also observed in Rice and Tse (1986) (the reference in L.298).
**[Answer] Your viewpoint is correct. In this study, I mainly focus on the effect on recurrence. The phase portrait is just used to express the possible change of fixed points due to either a use of different values of or a use of time-varying values of model parameters. Nonlinear behavior, including very small transient phase which was not observed in this study, of the system will be my next study.**

[25]L.309-310: I cannot understand that the right panels show $T_R$.
**[Answer] The related statements have been deleted in the revised manuscript.**

[26]L.314: I cannot understand why larger $\eta$ generates chaos.
**[Answer] The word "chaos" has been re-written as "an attractor" in the revised manuscript.**

[27]L.318: The slope values at V=0 and U=0 decrease with increasing $\eta$ more drastically for the larger $U_c$. As pointed out in my comment [2], the slope values should not be compared among the figures because $U_{max}$ and $V_{max}$ values are different among the figures.
**[Answer] As mentioned in my answer of your comment [2], for a certain figure we can the absolute values of slope in the four right-handed-sides panels because their values of $V_{max}$ and $U_{max}$ are the same. Of course, it is not good to compare the values in different figures due to different values of $V_{max}$ and $U_{max}$ in use.**

[28]L.319: The references are needed.

**[Answer] "The previous study" means the simulation results of this study. Hence, the words "The previous study" have been re-written as "The simulation results as mentioned previously".**

[29]L.321: "the effects" The effects of temporal variations of η and $U_c$?
**[Answer] The following statement "the effects of time-dependent η and $U_c$" has been added to the revised manuscript.**

[30]L.329-330: "Λ=($λ_f$-$λ_n$)/($β_f$+$β_n$)"
It would be better to move this to p.7, adding the definition of $λ_f$, $λ_n$, $β_f$, and $β_n$.
**[Answer] The statements have been re-written and added in the revised manuscript.**

[31]L.338: "$μ_f$" → "$μ_f$"
**[Answer] "$μ_f$" is replaced by "$μ_f$" in the revised manuscript.**

[32]L.347: "$ρ_f$" and "n" → "$ρ_f$" and "$n$"
**[Answer] "$ρ_f$" and "n" are replaced, respectively, by "$ρ_f$" and "$n$" in the revised manuscript.**

[33]L.362-364: The Author used the words "time-varying". However, "the increase in permeability can result in the increase in pore pressure due to slip" would be better because "This" in the sentence "This can reduce the frictional resistance" obviously means an increase in the pore pressure.
**[Answer] The statement "The time-varying permeability can result in the time-varying pore pressure, $p_f$" has been re-written as "An increase in permeability can result in an increase in pore pressure, $p_f$".**

[34]L. 410: "$\underline{C}$" → "$C$"?
**[Answer] "$\underline{C}$" is replaced by "$C$" in the revised manuscript.**

[35]L.411: "approaches unity" The slope values seem to become smaller than unity in Figs. 9-12. Plotting the slope values (with time or slip) may clarify this point.
Why does the unity important? The slope values depend on the $V_{max}$ and $U_{max}$ values.
**[Answer] Actually, the slope values become smaller than unity in Figs. 9-12. Hence, the statement "approach to unity" has been deleted in the revised manuscript.**

[36]Bizzarri (2010) showed the effects of the wear process on the stick-slip behaviors, assuming the friction law with thermal pressurization, and thus the results on the wear processes in this study are not new. I suggest that the statements on the results of the simulations including both the wear processes and the viscous effects (Figs. 11 and 12) are added.
**[Answer] I agree with you. Related information has been added to the revised manuscript.**

[37]Are the η and C values used in the simulations consistent with those estimated by observations or laboratory experiments in previous studies (e.g., Boneh et al., 2014, pageoph)?
**[Answer] This study is merely my first step to theoretically explore the earthquake recurrences caused by time-varying model parameters through numerical simulations. In this study, I just want to theoretical explore the possible effects on**

earthquake recurrences caused by time-varying of fault width. Hence, only the assumed values have been taken into account. I have not yet compared my values with those obtained by others. I will approach the problem for real faults in near future, and thus it is necessary to take the values of model parameters obtained from field observations and laboratory experiments into account.

[38] Vertical aces in Figs. 4-12: Please add the scales of the $\Sigma U/U_{max}$ aces. The maximum of $\Sigma U/U_{max}$ must be larger than 1 because $U/U_{max}$ reaches 1 or larger in the right panels of (a).
**[Answer] In Figs. 8–12, the velocity waveforms and displacements are normalized by the maximum values of each figure. Hence, the upper bound value of the vertical axis is 1. The statements have been added to the revised manuscript.**

[39] Fig.8-12: Why do the behaviors of the stick-slips (e.g., $T_R$, D, and $V_m$) vary with time in spite of the constant $U_c$?
**[Answer] The values of $U_c$ are not constant and vary with time in Figs. 8–12. The values of $U_c$ shown in the text and figure captions of have been re-written to be the initial values of $U_c$.**

[40] Figs.11(a) and 12(a): Why $V_m/V_{max} \neq 1$? I guessed that $V_{max}$ was defined as $V_m$ in (a) for each figure in Figs. 4-10. Why is the maximum of $U/U_{max}$ larger than 1? I guessed that $U_{max}$ was defined as the maximum of U in each figure in Figs. 4-10.
**[Answer] It is very sorry that the numerical computation cannot work when C is larger than a certain value which depends on $U_c$. The normalization of original Fig. 11 and Fig 12 was made based on the maximum values of panel (d), which are wrong. The re-computed results are displayed in the revised manuscript.**

[41] Figs. 11(d) and 12(d): Why are there two thin solid lines? Why is $\Sigma U/U_{max}$ constant (thick solid line)?
**[Answer] It is very sorry that the numerical computation cannot work when C is larger than a certain value. Originally, I kept the bad simulation results for C=0.05 in Figs. 11–12 with $\eta=1$, because I wanted to retain the same four values of C for Figs. 8–12. This idea sounds not good. Hence, I have re-done the numerical computations to find out the upper bound value of C for a certain $U_c$ with $\eta=1$. The re-computed results are displayed in Figs. 11–12 of the revised manuscript.**